

# A web of 2d dualities: $Z_2$ gauge fields and Arf invariants

**Andreas Karch**[1⋆]**, David Tong**[2†]** and Carl Turner**[2‡]

**1** Department of Physics, University of Washington, Seattle, WA 98195, USA
**2** Department of Applied Mathematics and Theoretical Physics,
University of Cambridge, Cambridge, CB3 OWA, UK

⋆ akarch@uw.edu, † d.tong@damtp.cam.ac.uk, ‡ C.P.Turner@damtp.cam.ac.uk

## Abstract

We describe a web of well-known dualities connecting quantum field theories in d=1+1 dimensions. The web is constructed by gauging $Z_2$ global symmetries and includes a number of perennial favourites such as the Jordan-Wigner transformation, Kramers-Wannier duality, bosonization of a Dirac fermion, and T-duality. There are also less-loved examples, such as non-modular invariant c=1 CFTs that depend on a background spin structure.

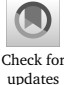
# 1 Introduction

Dualities have a tendency to proliferate. Given one duality, which relates seemingly different quantum field theories, it is often possible to deform both sides in a controlled manner and infer the existence of further dualities. This can be achieved in an obvious fashion, by adding relevant operators to both sides, or by doing something more drastic such as promoting global symmetries to gauge symmetries, or compactifying to reduce the spacetime dimension in which the theories live. In this manner, a single seed duality can give rise to a web of further dualities.

A good example of this can be found in recent developments in $d = 2 + 1$ dimensional gauge theories [1, 2]. There, the seed is a 3d bosonization duality in which scalars coupled to Chern-Simons terms have a simple fermionic description. This duality is similar in spirit, but different in details, to one suggested long ago by Polyakov [3]. A precise statement of the duality, with accompanying compelling evidence, came only after input from condensed matter physics [4, 5], higher spin holography [6–9], and supersymmetry [10–14]. With the seed duality in place, one can build further dualities by playing with background Chern-Simons terms and gauging various $U(1)$ symmetries, a methodology previously advocated in [15–17]. The resulting web of dualities includes some familiar faces like particle-vortex duality [18–20] together with more recently discovered dualities, such as the fermionic particle-vortex duality proposed in [21–23] and the self-duality of [24, 25]. (The fermionic particle-vortex duality was, in large part, the motivation to delve deeper into the web connecting different dualities which, in turn, offers a more precise statement about how the two fermionic theories are related [2, 26].) There has been a great deal of further work in this area, including [29–42]. An excellent review of many of these developments can be found in [43].

The purpose of this paper is to describe a similar web of dualities in $d = 1 + 1$ dimensions. Instead of $U(1)$ gauge fields, the web is constructed by gauging various $\mathbf{Z}_2$ symmetries. The role of the Chern-Simons terms in $d = 2 + 1$ dimensions is played by an object known as the *Arf invariant* in $d = 1 + 1$. As we will review in some detail in Section 2, this is a mod 2 invariant built from $\mathbf{Z}_2$ gauge connections and an underlying spin structure. The Arf invariant also arises in the guise of the mod 2 index of the Dirac operator.

Neither the dualities nor the methodology in this paper are new. Our seed will be provided by a continuum version of the bosonization of Jordan and Wigner, a duality first discovered on the lattice in 1928 [44]. Further down the web, we will find Kramers-Wannier duality emerging. This is more modern, dating from 1941 [45]. As we progress we will see other perennial favourites, including Coleman's bosonization of a Dirac fermion [46] and T-duality, albeit written in a somewhat unfamiliar presentation.

The basic manipulations of $\mathbf{Z}_2$ gauge fields and Arf invariants which allow one to build the duality web were first described in abridged form in important papers by Kapustin, Thorngren, Turzillo and Wang [47, 48]. The key idea – that one can start from Jordan-Wigner bosonization of a Majorana fermion, and subsequently derive Kramers-Wannier duality – is already contained in these papers. A brief summary of these results has appeared in the review article [43][1]. A lattice version of the 2d duality web was recently proposed in [49]; our results here can be viewed as the continuum limit of these lattice dualities. More recent, related developments include [50, 51].

Although the basics of the 2d duality web have appeared previously, it is a beautiful story and one which, in our opinion, deserves exposure to a wider audience with the details fleshed out. This is our goal in Section 2. We follow this up in Section 3 by extending these ideas

---

[1]After this work was completed, we found essentially identical ideas in an entertaining lecture by Yuji Tachikawa: https://www.youtube.com/watch?v=HZEIk9ucr9Q.

from Majorana to Dirac fermions. We will be able to reproduce a number of well-known familiar results about bosonization and $c = 1$ conformal field theories. We will also explicitly map out the space of $c = 1$ CFTs that depend on a spin structure. Of course, the study of $c = 1$ CFTs is a very well trodden path. Nonetheless, we believe there is value in walking this path in unfamiliar shoes, and seeing connections that were not previously apparent. We also include three appendices: one where we describe a number of properties of the Arf invariant, one reviewing the partition function of a compact boson, and one clarifying how the discrete gauge fields together with their topological terms can arise from compactification from 2+1 dimensions.

## 2  Majorana Fermions and the Ising Model

Dualities beget further dualities. The purpose of this section is to explore this begetting, starting from the relationship between a Majorana fermion and the Ising model in $d = 1 + 1$ dimensions.

### 2.1  The Majorana Fermion

Before we introduce the dualities, we need a few simple facts about the life of a Majorana fermion in two dimensions. For the most part, we work in Euclidean signature on a general Riemann surface $X$. We take the gamma matrices to be $\gamma^1 = \sigma^1$ and $\gamma^2 = \sigma^3$ so that $\gamma^3 = \gamma^1 \gamma^2 = -i\sigma^2$ is real and anti-symmetric.[2]

To specify a theory of fermions requires a choice of spin structure on $X$. For our purposes, this spin structure tells us whether the fermions are periodic or anti-periodic around each cycle. We denote this spin structure by $\rho$. This will play an important role in what follows, so we write the Dirac operator as $\slashed{D}_\rho$, with the subscript labelling the choice of spin structure.

The action for a massive Majorana spinor is

$$S_{\text{Maj}} = \int_X i\bar{\chi}\slashed{D}_\rho\chi + im\bar{\chi}\gamma^3\chi. \tag{2.1}$$

The Majorana fermion has two distinct phases, depending on the sign of the mass. The clearest physical manifestation of this difference arises when we consider the theory on a Lorentzian signature spacetime with time-like boundary, a situation which is the continuum limit of the Kitaev Majorana chain [52]. For a given choice of boundary condition, one finds a fermion zero mode localised on the boundary for one sign of the mass, but not the other.

Here we are interested on manifolds $X$ without boundary. Nonetheless, there is a subtle remnant of the topological phase. The partition function for the Majorana fermion is

$$Z_{\text{Maj}}[\rho; m] = \text{Pf}(\slashed{D}_\rho + m\gamma^3).$$

The Pfaffian of the Dirac operator is real. However, there is no natural choice of sign. For a given $m > 0$, we are at liberty to define the sign to be positive. The question then becomes: can the sign of the Pfaffian change as we vary $m$?

---

[2]Later, we will also have cause to work in Lorentzian signature. Here the gamma matrices are $\gamma^0 = i\sigma^2$ and $\gamma^1 = \sigma^1$, so that $\gamma^3 = \gamma^0\gamma^1 = \sigma^3$. Spinor conjugation in Euclidean signature is $\bar{\chi} = \chi^\dagger$, and in Lorentzian signature it is $\bar{\chi} = \chi^\dagger\gamma^0$.

The Pfaffian changes sign when eigenvalues of the Dirac operator cross zero. This can only occur when $m = 0$ where the massless Dirac operator $\slashed{D}_\rho$ may have zero modes. Such zero modes always come in chiral pairs, labelled by the eigenvalue $\pm 1$ under the chiral operator $i\gamma^3$. Restricted to a pair of zero modes, we have

$$\text{Pf}\big(\slashed{D}_\rho + m\gamma^3\big)\Big|_{\text{zero mode}} = \text{Pf}\begin{pmatrix} 0 & -m \\ m & 0 \end{pmatrix} = m,$$

which clearly changes sign as $m$ varies from positive to negative.

We learn that our partition function does indeed change sign whenever there are an odd number of pairs of zero modes of $\slashed{D}_\rho$. This, in turn, depends on the spin structure $\rho$ and is determined by the mod 2 index of the Dirac operator, restricted to modes of a given chirality. We denote this as

$$\mathcal{I}[\rho] = \text{Index}(\slashed{D}_\rho) \in \mathbf{Z}_2.$$

We will discuss more properties of this mod 2 index shortly. For now, it suffices to point out that for spin structures for which $\mathcal{I}(\rho) = 1$, the partition function has a different sign for $m > 0$ and $m < 0$ [53],

$$Z_{\text{Maj}}[\rho; -m] = (-1)^{\mathcal{I}[\rho]} Z_{\text{Maj}}[\rho; m]. \tag{2.2}$$

In the context of the Kitaev Majorana chain, this relation dates back to [47] (see also [54]) where the mod 2 index is replaced by an equivalent object known as the Arf invariant. The relation between these will be elaborated upon below.

The transformation (2.2) can be viewed as an anomaly in the discrete chiral symmetry. This takes slightly different forms in Euclidean and Lorentzian signature[3]:

$$\mathbf{Z}_2 : \begin{cases} \chi \mapsto i\gamma^3 \chi & \text{Euclidean} \\ \chi \mapsto \gamma^3 \chi & \text{Lorentzian}. \end{cases}$$

This $\mathbf{Z}_2$ chiral transformation leaves the kinetic term in (2.1) invariant, but flips the sign of the mass and therefore maps $\mathbf{Z}_2 : Z_{\text{Maj}}[\rho, m] \mapsto Z_{\text{Maj}}[\rho, -m]$. The transformation (2.2) tells us that, for certain spin structures $\rho$, even the partition function for a massless Majorana fermion may not be invariant under the chiral transformation. This is the sense in which it is anomalous. In a slight abuse of notation, we will say that the effective Euclidean action transforms under the discrete chiral symmetry as

$$\mathbf{Z}_2 : \int_X i\bar{\chi}\slashed{D}_\rho \chi \mapsto \int_X i\bar{\chi}\slashed{D}_\rho \chi + i\pi \mathcal{I}[\rho]. \tag{2.3}$$

## 2.2 Properties of the Mod 2 Index

The mod 2 index of the chiral Dirac operator will play a key role in what follows and it is useful to review a few of its properties. As we described above, the spin structure $\rho$ specifies whether fermions are periodic (P) or anti-periodic (AP) around a given cycle. (In string theory, these are referred to as Ramond and Neveu-Schwarz boundary conditions respectively.) The

---

[3]The extra factor of $i$ in Euclidean signature may look strange since it does not respect the reality of the Majorana fermion. However, as explained in [53], the reality conditions on fermions and their symmetries should be imposed in Lorentzian signature, where the chiral symmetry is quite sensible. Upon Wick rotation, we pick up an extra factor of $i$ from $\gamma^0$. Indeed, without the factor of $i$ in Euclidean space, the kinetic term in the action changes sign under this symmetry.

number of different, inequivalent spin structures is given by $|H_1(X, \mathbf{Z}_2)|$ which, for us, is $2^{2g}$ with $g$ the genus of $X$.

For example, when $X = \mathbf{T}^2$, there are four inequivalent spin structures, given by a choice of P or AP around each of the two cycles. Choosing the flat metric on the torus, a Majorana zero mode of the Dirac operator is simply a constant spinor. This is admissible only when the spin structure has periodic boundary conditions around both cycles. The mod 2 index is then given by

$$\mathcal{I}[\rho] = \begin{cases} 1 & \rho = PP \\ 0 & \rho = AP, PA, AA \end{cases}. \tag{2.4}$$

More generally, on a Riemann surface of genus $g$, there are $2^{g-1}(2^g - 1)$ spin structures which have an odd number of zero modes (typically one) for which $\mathcal{I}[\rho] = 1$, and there are $2^{g-1}(2^g + 1)$ spin structures which have an even number of zero modes (typically none) for which $\mathcal{I}[\rho] = 0$. This follows from the fact that the number of chiral zero modes mod 2 is a bordism invariant, meaning that it is additive when we glue together Riemann surfaces. So, for example, we can construct a $g = 2$ Riemann surface with $\mathcal{I}[\rho] = 0$ by gluing together two $g = 1$ Riemann surfaces, both of which have the same value of $\mathcal{I}[\rho]$. There are $1 \times 1 + 3 \times 3 = 10$ ways of doing this. A pedagogical physics discussion of these issues can be found in [55].

Our real interest in this paper is in matter – both fermions and scalars – coupled to $\mathbf{Z}_2$ gauge fields $s \in H^1(X, \mathbf{Z}_2)$. These $\mathbf{Z}_2$ gauge fields are specified by the holonomy around each cycle of $X$. (We will also discuss disorder operator in these theories which can be viewed as inserting $\mathbf{Z}_2$ flux.)

There is close relationship between $\mathbf{Z}_2$ gauge fields and spin structures. As we have seen, the latter already determine whether a fermion is periodic or anti-periodic around a given cycle $\gamma$. Meanwhile, a $\mathbf{Z}_2$ gauge connection $s$ has holonomy $\int_\gamma s \in \{0, 1\}$ around each cycle. When $\int_\gamma s = 0$ this does nothing, but when $\int_\gamma s = 1$, the holonomy shifts the boundary conditions from periodic to anti-periodic, and vice versa. This means that, given a spin structure $\rho$ and a $\mathbf{Z}_2$ gauge connection $s$, we can construct a new spin connection which we denote as $s \cdot \rho$. The mod 2 index of the new spin structure $\mathcal{I}[s \cdot \rho]$ obeys a number of useful properties:

**Claim 1:** The first property arises when we combine $\mathbf{Z}_2$ gauge fields. We have

$$\mathcal{I}[(s + t) \cdot \rho] = \mathcal{I}[s \cdot \rho] + \mathcal{I}[t \cdot \rho] + \mathcal{I}[\rho] + \int s \cup t, \tag{2.5}$$

where the equality holds mod 2. (No harm will come to you if you prefer to think of the cup product as $\int s \wedge S$.) A proof of this identity can be found in [56].

Algebraically, the expression (2.5) looks very much like a quadratic function on $H^1(X, \mathbf{Z}_2)$, with the cup product playing the role of the cross-term. Indeed, such a function is sometimes referred to as a quadratic refinement of the cup product. This underlies the fact that the mod 2 index can be identified as a quadratic invariant of $\rho$ known as the *Arf invariant* [56]:

$$\mathcal{I}[\rho] = \text{Arf}[\rho]. \tag{2.6}$$

More details on this relation can be found in the Appendix A. For the purposes of this paper, we will use the notation $\mathcal{I}[\rho]$ and $\text{Arf}[\rho]$ interchangeably. In particular, when discussing dualities below we use the notation $\text{Arf}[\rho]$, following the usage in earlier papers on the subject [43, 47, 48, 54].

The second result involves summing over all possible $\mathbf{Z}_2$ gauge fields. When the background space $X$ has genus $g$, the correct normalisation of the path integral for the sum over a $\mathbf{Z}_2$ gauge field $s$ is

$$\frac{1}{2^g}\sum_s .$$

To see this, first note that if a $\mathbf{Z}_2$ gauge field appears linearly in the path integral, then it acts as a Lagrange multiplier, with

$$\frac{1}{2^g}\sum_s (-1)^{\int s\cup t} = \begin{cases} 2^g & \text{if } t=0 \\ 0 & \text{otherwise} \end{cases} . \tag{2.7}$$

The normalisation of $1/2^g$ then ensures that the trivial theory,

$$\frac{1}{2^g}\sum_s \frac{1}{2^g}\sum_t (-1)^{\int s\cup t} = 1,$$

is indeed trivial. The second result that we need is then

**Claim 2:**

$$\frac{1}{2^g}\sum_s (-1)^{\mathcal{I}(s\cdot\rho)+\mathcal{I}[\rho]+\int s\cup t} = (-1)^{\mathcal{I}[t\cdot\rho]}. \tag{2.8}$$

To show this, we start by noting that

$$\sum_s (-1)^{\mathcal{I}[s\cdot\rho]} = 2^{g-1}(2^g+1) - 2^{g-1}(2^g-1) = 2^g.$$

But, from Claim 1, we also have

$$\sum_s (-1)^{\mathcal{I}[s\cdot\rho]} = \sum_s (-1)^{\mathcal{I}[(s+t)\cdot\rho]} = (-1)^{\mathcal{I}[t\cdot\rho]}\sum_s (-1)^{\mathcal{I}[s\cdot\rho]+\mathcal{I}[\rho]+\int s\cup t}.$$

Combining these yields the desired result. Both (2.5) and (2.8) will be invoked frequently in what follows.

## 2.3 Majorana = Ising/$\mathbf{Z}_2$

We now turn to the main topic of the paper: dualities. We will construct a number of dualities which relate bosonic and fermionic matter, coupled to $\mathbf{Z}_2$ gauge fields. We will adopt the convention that lower-case $\mathbf{Z}_2$ connections, such as $s$ and $t$, are dynamical, while upper-case $\mathbf{Z}_2$ connections, such as $S$ and $T$, are background.

We start with a seed duality. Roughly speaking, this is the equivalence between a single Majorana fermion and the Ising model. Here the "roughly speaking" refers to the way that various $\mathbf{Z}_2$ gauge fields appear and will be at the heart of our story. The duality can be traced back to the Jordan-Wigner transformation [44], which is a change of variables that provides rather simple solutions to a number of 2d spin systems, including the Ising model [58, 59]. In the continuum, this duality takes a rather more subtle form, as first explained in [47] (see also [43]) and can be schematically written as

$$\int i\bar{\chi}\slashed{D}_{S\cdot\rho}\chi \quad\longleftrightarrow\quad \int (\mathcal{D}_s\sigma)^2 + \sigma^4 + i\pi\Big[\mathrm{Arf}[s\cdot\rho] + \mathrm{Arf}[\rho] + \int s\cup S\Big]. \tag{2.9}$$

We have introduced two $\mathbf{Z}_2$ gauge connections: the fermion is coupled to a background gauge connection $S$, while the scalar is coupled to a dynamical $\mathbf{Z}_2$ gauge connection $s$. These are associated to the respective $\mathbf{Z}_2$ symmetries

$$\mathbf{Z}_2^S : \chi \mapsto -\chi \quad \text{and} \quad \mathbf{Z}_2^s : \sigma \mapsto -\sigma.$$

Note that $\mathbf{Z}_2^S$ is better known as $(-1)^F$ and coincides with a $2\pi$ rotation in space. On the scalar side of the theory, these gauge fields are coupled together through the cup product. The $\sigma^4$ coupling on the right-hand-side should be taken to mean that we flow to the Ising fixed point, and subsequently gauge the $\mathbf{Z}_2$ symmetry.

Here is an obvious point: both sides of the duality, including the scalar theory, require the existence of a spin structure $\rho$ in order to be defined. In this sense, the right-hand side is, despite appearances, not a bosonic quantum field theory.

**Matching Phases**

We will now proceed to explore various aspects of the duality. This will allow us to better understand the role played by the $\text{Arf}[s \cdot \rho] + \text{Arf}[\rho]$ term and cup product terms in the scalar theory.

We start by studying the phases of the two sides. To do this, we deform away from the fixed point by adding a mass $m$ for the fermion, as in (2.1). We expect this to be dual to a mass $M^2\sigma^2$ for the boson.

We described the partition function for the fermion in the previous section. The theory lies in a trivial, gapped phase when $m > 0$. In contrast, when $m < 0$ the theory lies in a topological phase. Upon integrating out the fermion, this is seen by the effective action (2.2)

$$S_{\text{eff}} = i\pi \text{Arf}[S \cdot \rho]. \tag{2.10}$$

The fact that a Majorana fermion in the non-trivial phase of the Kitaev chain has an effective action given by the Arf invariant (or, equivalently, the mod 2 index) was first explained in [47], and was elaborated upon in [54]. As stressed in [43], this is reminiscent of the manner in which Chern-Simons terms are generated in three dimensions depending on the sign of the fermion mass. Indeed, the analogy between the Arf invariant and Chern-Simons terms will develop further as we go along.

Now we can match this to the bosonic theory. When we take $M^2 > 0$, we can simply integrate out the scalar to leave ourselves with the theory of the dynamical $\mathbf{Z}_2$ gauge field,

$$Z_{\text{scalar}} = \sum_s \exp\left(i\pi\left[\text{Arf}[s \cdot \rho] + \text{Arf}[\rho] + \int s \cup S\right]\right) \sim \exp\left(i\pi \text{Arf}[S \cdot \rho]\right), \tag{2.11}$$

where, in the final equality, we used the relation (2.8). Note that this coincides with the low-energy fermionic theory (2.10) when $m < 0$.

In contrast, when $M^2 < 0$, the scalar condenses and breaks the $\mathbf{Z}_2$ gauge symmetry, ensuring that $s = 0$ in the ground state. In this case, $\text{Arf}[s \cdot \rho] + \text{Arf}[\rho] = 0$ (recall, the Arf invariant is defined mod 2) and so the low-energy effective action is independent of the fiducial spin structure and background field $S$. This coincides with the trivial fermionic theory $m > 0$.

We see that the two phases match if the fermionic and bosonic masses are related by

$$m \quad \longleftrightarrow \quad -M^2. \tag{2.12}$$

**Matching States**

It is also useful to understand how the Hilbert spaces of the two theories map into each other. For this, we rotate to Lorentzian signature and work on $X = \mathbf{R} \times \mathbf{S}^1$. The duality should hold for any choice of the fiducial spin structure $\rho$ which, for us, is now the question of whether we have periodic or anti-periodic boundary conditions around the spatial $\mathbf{S}^1$. We'll see how this works.

First, consider anti-periodic boundary conditions $\rho$. We will also start by setting the background $\mathbf{Z}_2$ connection $S = 0$. This is the Neveu-Schwarz, or twisted, sector of the fermion. On the Ising side, the $\mathbf{Z}_2$ gauge field is dynamical, which means that we must sum over both twisted and untwisted sectors. In the untwisted sector, the gauge field restricts us to excitations that are even under $\mathbf{Z}_2$. However, the presence of the Arf invariant $\mathrm{Arf}[s \cdot \rho]$ gives a $\mathbf{Z}_2$ charge to the twisted sector. Gauge invariance means that we must then excite an odd numbers of $\sigma$ excitations. In other words, the Hilbert spaces on the two sides of the duality are matched as [48]

$$\mathcal{F}_{NS} = \mathcal{B}_R^+ \oplus \mathcal{B}_{NS}^-, \tag{2.13}$$

where the $\pm$ refers to the even/odd sectors under the gauged $\mathbf{Z}_2$, and we have adopted the fermionic notation for the bosons, referring to the untwisted sector as R, and the twisted sector as NS. Note that this, and subsequent statements, are equalities of the spectrum of the Hamiltonian on these two on Hilbert spaces.

What happens if we turn on the background $\mathbf{Z}_2$ gauge field, $\int_{\mathbf{S}^1} S = 1$? The fermion now sits in Ramond, or untwisted, sector. In the bosonic theory, the role of $\int s \cup S$ term is to provide an extra $\mathbf{Z}_2$ charge to the system, so that the Hilbert space consists of states in the untwisted sector that are odd under $\mathbf{Z}_2$, and states in the twisted sector that are even. We now have [48]

$$\mathcal{F}_R = \mathcal{B}_R^- \oplus \mathcal{B}_{NS}^+. \tag{2.14}$$

There is an interesting story lurking here. The Hamiltonian on $\mathcal{F}_R$ is two-fold degenerate. This is because a Majorana spinor on $X = \mathbf{R} \times \mathbf{S}^1$ has a single real zero mode (a constant spinor) from which we can form a single, complex zero mode $\psi = \chi_L + i \chi_R$ with $\chi_{L/R}$ chiral fermions. This zero mode provides the degeneracy of the spectrum, with states distinguished by their charge under $(-1)^F = \mathbf{Z}_2^S$. On the bosonic side, this degeneracy is not manifest but, as we will see in Section 2.5, actually arises because

$$\mathcal{B}_R^- = \mathcal{B}_{NS}^+, \tag{2.15}$$

again with the equality implying equivalence of the spectra.

The degeneracy of the spectra also provides an explanation for the anomaly in the discrete chiral transformation (2.3). The chiral transformation, $\chi \mapsto \gamma^3 \chi$ decomposes as $\chi_L \mapsto \chi_L$ and $\chi_R \mapsto -\chi_R$, so the zero mode $\psi \mapsto \psi^\dagger$. This, in turn, exchanges the $\mathbf{Z}_2^S = (-1)^F$ charge of the states. Correspondingly, on the bosonic side the chiral transformation must exchange $\mathcal{B}_R^-$ and $\mathcal{B}_{NS}^+$; we will see how this works in Section 2.5.

Suppose that we instead choose the fiducial background spin structure $\rho$ to be periodic. Now, with $S = 0$, the fermionic theory has the Ramond Hilbert space $\mathcal{F}_R$. On the Ising side, the $\mathrm{Arf}[s \cdot \rho]$ term endows the untwisted sector with a $\mathbf{Z}_2$ charge, a role previously played by $\int s \cup S$. Meanwhile, for $S \neq 0$ we have the Neveu-Schwarz sector, and the combination of $\mathrm{Arf}[s \cdot \rho] + \int s \cup S$ ensure that the twisted sector on the Ising side is $\mathbf{Z}_2$ odd. We again find the decomposition (2.13) and (2.14).

**Matching Operators**

The discussion above can also be phrased in terms of local operators. Specifically, we are interested in operators which carry fermion number $\mathbf{Z}_2^S = (-1)^F$. The cup product $\int s \cup S$ on the scalar side of (2.9) shows that such operators are necessarily kinks in the scalar field. These are described by disorder operators $\mu(x)$, which can be thought of as an instanton in the dynamical $s$ field. Such operators create states in $\mathcal{B}_{NS}$.

However, there is a further subtlety that arises when we take $\rho$ to describe anti-periodic boundary conditions on $\mathbf{S}^1$. This is due to the presence of the $\mathcal{I}[s \cdot \rho]$ term which then endows a kink field with $\mathbf{Z}_2$ gauge charge. This means that to construct a gauge invariant state, the kink operator must be dressed with a further $\mathbf{Z}_2$ gauge charge carried by $\sigma(x)$. This can be seen in the identification of Hilbert spaces (2.13) where the kink sector carries $\mathbf{Z}_2$ gauge charge +1. Schematically, we have the map between local operators

$$\chi(x) \quad \longleftrightarrow \quad \mu(x)\sigma(x). \tag{2.16}$$

There is a close analogy here to the story of 3d bosonization [43]. In that case, a Chern-Simons term of the form $\int a \mathrm{d}a$ endows the monopole operator with $U(1)$ gauge charge, which must subsequently be cancelled by dressing the monopole with appropriate matter excitations. In this way, the Chern-Simons term plays the same role as the Arf invariant in the duality which forces the kink operator to be similarly dressed. This analogy also stretches to the cup product $\int s \cup S$, which plays the role of the BF coupling $\int a \mathrm{d}A$ in 3d.

## 2.4   Majorana/$\mathbf{Z}_2$ = Ising

In three dimensional quantum field theories, the existence of a seed bosonization duality allowed for the construction of a web of further dualities, including both bosonic and fermionic particle vortex duality [1, 2]. This was accomplished by adding background Chern-Simons terms and subsequently promoting background fields to become dynamical.

One can play the same game with the 2d dualities and their $\mathbf{Z}_2$ gauge fields, a point first stressed in [43]. To this end, we couple the background gauge field $S$ to a second background gauge field $T$, and subsequently promote $S$ to become dynamical. The duality (2.9) then becomes

$$\int i\bar{\chi}\slashed{D}_{t\cdot\rho}\chi + i\pi\int t \cup T \quad \longleftrightarrow \quad \int (\mathcal{D}_s\sigma)^2 + \sigma^4 + i\pi\Big[\mathrm{Arf}[s\cdot\rho] + \mathrm{Arf}[\rho] + \int (s+T)\cup t\Big],$$

where we don't care about minus signs arising in the ordering of the cup product because it's defined mod 2. On the right-hand-side, the newly dynamical gauge field $t$ appears only linearly and so, using (2.7), acts as a Lagrange multiplier setting $s = T$ mod 2, leaving us with

$$\int i\bar{\chi}\slashed{D}_{t\cdot\rho}\chi + \pi\int t \cup T \quad \longleftrightarrow \quad \int (\mathcal{D}_T\sigma)^2 + \sigma^4 + i\pi\Big[\mathrm{Arf}[T\cdot\rho] + \mathrm{Arf}[\rho]\Big].$$

This is now a duality between the Ising model and a $\mathbf{Z}_2$ gauge theory coupled to a Majorana fermion. The Arf invariants involve only the background fields, so we are at liberty to take them over to the other side of the duality. Renaming some of the gauge fields, we have

$$\int i\bar{\chi}\slashed{D}_{s\cdot\rho}\chi + i\pi\Big[\mathrm{Arf}[S\cdot\rho] + \mathrm{Arf}[\rho] + \int s\cup S\Big] \quad \longleftrightarrow \quad \int (\mathcal{D}_S\sigma)^2 + \sigma^4. \tag{2.17}$$

Note that, despite appearances, the fermionic theory does not depend on the choice of fiducial spin structure $\rho$. To see this, write $\rho = R \cdot \rho'$ for some $\mathbf{Z}_2$ connection $R$ and spin structure

$\rho'$. Then a few manipulations show that the left-hand-side of (2.17) takes the same form, but with $\rho$ replaced by $\rho'$.

We can once again match both phases and Hilbert spaces. First, the phases. The theory on the right hand side is the Ising model. It has two phases as we vary the mass $M^2\sigma^2$, but neither of them are topological. Instead the two phases are distinguished in the infinite volume limit in the usual Landau fashion by the symmetry $\mathbf{Z}_2$.

We can see how this is matched in the duality. Turn on a mass $m$ for the fermion, and integrate it out. For $m < 0$ we generate an extra term in the low-energy effective action, $\text{Arf}[s \cdot \rho]$. Summing over the holonomies of $s$ transforms this into $\text{Arf}[S \cdot \rho]$ using (2.8), but we already have such a term on the left hand side and $2\text{Arf}[S \cdot \rho] = 0$. (Because two 'arfs make an 'ole.) We learn that for $m < 0$ we sit in the trivial phase.

What about $m > 0$? After integrating out the fermion, we are left with the $\int s \cup S$ term in the effective action. For $S = 0$ the sum over $s$ gives rise to the two ground states seen on the bosonic side in the infinite volume limit. Meanwhile, when $S \neq 0$, the cup product requires that gauge invariant states must have an odd number of fermions excited. In other words, the vacuum has energy $\sim m$. This matches the $\mathbf{Z}_2$ broken phase of the Ising model where, for $S \neq 0$, we sit in the twisted sector and the ground state corresponds to the domain wall. Once again, we find the map $m \longleftrightarrow -M^2$ between the masses on the two sides of the duality.

We can make the matching between states more precise by considering the theory on $X = \mathbf{R} \times \mathbf{S}^1$. When $S = 0$ we have the usual Ising model in the untwisted sector. What are the corresponding states in the fermionic side? The dynamical $\mathbf{Z}_2$ gauge field allows for either periodic or anti-periodic boundary conditions. However, in the absence of the Arf invariant coupling for the dynamical gauge field, both sectors have even fermion parity. We have that [48]

$$\mathcal{B}_R = \mathcal{F}_R^+ \oplus \mathcal{F}_{NS}^+.$$

Meanwhile, when $S \neq 0$, we have the twisted sector of the Ising model. In the fermionic theory, the $\int s \cup S$ term obliges us to excite a single fermion, giving [48]

$$\mathcal{B}_{NS} = \mathcal{F}_R^- \oplus \mathcal{F}_{NS}^-.$$

**The Chiral Transformation Revisited**

We already met the chiral transformation in Section 2.1. This flips the sign of the fermion mass. At the critical point, the chiral transformation acts on the fermion as (2.3)

$$\int_X i\bar{\chi} \slashed{D}_{S \cdot \rho}\, \chi \; \mapsto \; \int_X i\bar{\chi} \slashed{D}_{S \cdot \rho}\, \chi + i\pi\, \text{Arf}[S \cdot \rho]. \tag{2.18}$$

We can ask: how does the chiral transformation act on our dualities?

For the Majorana = Ising/$\mathbf{Z}_2$ duality of Section 2.3, the chiral transformation adds the Arf invariant for a background field, $\text{Arf}[S \cdot \rho]$. This does not affect the dynamics of the theory. Nonetheless, it is interesting to ask how this is reproduced on the Ising side, which would appear to remain invariant under the chiral transformation. We will postpone the answer to this question until Section 2.5.

In contrast, if the fermion is coupled to a dynamical gauge field $s$ then the chiral transformation adds $\text{Arf}[s \cdot \rho]$ to the fermionic theory, and would appear to change its dynamics. Yet

there is no accompanying transformation on the scalar side. Acting on the duality (2.17), we can construct a different version of Majorana /$\mathbf{Z}_2$ = Ising duality,

$$\int i\bar{\chi}\slashed{D}_{s\cdot\rho}\chi + i\pi\left[\text{Arf}[s\cdot\rho]+\text{Arf}[S\cdot\rho]+\text{Arf}[\rho]+\int s\cup S\right] \quad\longleftrightarrow\quad \int (\mathcal{D}_S\sigma)^2 + \sigma^4. \quad (2.19)$$

Let's again perform some sanity checks to see how this duality pans out. Turn on a mass $m'$ for the fermion. This time the theory is the trivial phase when $m' > 0$, and in the $\mathbf{Z}_2$ broken phase when $m' < 0$. In other words, the new duality (2.19) has the map

$$m' \quad\longleftrightarrow\quad +M^2,$$

which is consistent with the idea that the chiral transformation flips the sign of the fermion mass.

In this new duality, the matching of Hilbert spaces differs. When $S = 0$, Hilbert space of the Ising model sits in the untwisted sector. The dynamical $\mathbf{Z}_2$ gauge field for the fermion allows both Ramond and Neveu-Schwarz boundary conditions, but the presence of the Arf term means that they come with differing fermion parity. Regardless of the fiducial spin structure $\rho$, we have

$$\mathcal{B}_R = \mathcal{F}_R^- \oplus \mathcal{F}_{NS}^+.$$

Meanwhile, when $S \neq 0$, we have the twisted sector of the Ising model. In the fermionic theory, the $\int s \cup S$ term obliges us to excite a single fermion, giving us

$$\mathcal{B}_{NS} = \mathcal{F}_R^+ \oplus \mathcal{F}_{NS}^-.$$

The fermionic parity in the Ramond sector is flipped relative to our earlier duality. (This option was also noted in a footnote in [48].)

## 2.5 Kramers-Wannier Duality

We can combine the bosonization dualities above to derive a purely bosonic duality. We work with the duality in the form (2.17) and again promote the background field $S$ to a dynamical field which we call $t$. We have

$$\int i\bar{\chi}\slashed{D}_{s\cdot\rho}\chi + i\pi\left[\text{Arf}[t\cdot\rho]+\text{Arf}[\rho]+\int (s+T)\cup t\right] \quad\longleftrightarrow\quad \int (\mathcal{D}_t\sigma)^2 + \sigma^4 + i\pi\int t\cup T.$$

On the left-hand side, we use the expression (2.8) to get

$$\int i\bar{\chi}\slashed{D}_{s\cdot\rho}\chi + i\pi\,\text{Arf}[(s+T)\cdot\rho] \quad\longleftrightarrow\quad \int (\mathcal{D}_t\sigma)^2 + \sigma^4 + i\pi\int t\cup T.$$

At this point we use the identity (2.5) for the Arf invariant, giving the duality

$$i\bar{\chi}\slashed{D}_{s\cdot\rho}\chi + i\pi\left[\text{Arf}[s\cdot\rho]+\text{Arf}[T\cdot\rho]+\text{Arf}[\rho]+\int s\cup T\right] \quad\longleftrightarrow\quad (\mathcal{D}_t\sigma)^2 + \sigma^4 + i\pi\int t\cup T.$$

Note that the right hand side is not a theory that we've previously encountered: it is Ising/$\mathbf{Z}_2$ but, in contrast to our duality (2.9) there is no Arf[$s\cdot\rho$] for the dynamical gauge field. Meanwhile, on the left-hand side we have a sector that looks like Majorana/$\mathbf{Z}_2$ with an Arf invariant

for the dynamical field. But this is precisely the form that appears in the chirally-transformed duality (2.19). Invoking this gives the scalar-scalar duality

$$\int (\mathcal{D}_S \sigma)^2 + \sigma^4 \quad \longleftrightarrow \quad \int (\mathcal{D}_t \tilde{\sigma})^2 + \tilde{\sigma}^4 + i\pi \int t \cup S. \tag{2.20}$$

This is Kramers-Wannier duality.

The derivation above closely mimics that of 3d particle-vortex duality from bosonization [1,2]. In the 3d case, the bosonization duality had a hidden time-reversal invariance; in the present case that role is played by the discrete chiral transformation. The relationship between the Jordan-Wigner transformation and Kramers-Wannier duality was previously stressed (with a slightly different logic) in [43].

We can play the same games that we saw previously and try to match states on $\mathbf{R} \times \mathbf{S}^1$. The story on the left-hand side is clear. When $M^2 > 0$ the $\mathbf{Z}_2$ global symmetry is intact. The Hilbert space of the theory comes from either the untwisted sector (when $S = 0$) or the twisted sector (when $S = 1$). In contrast, with $M^2 < 0$ the global $\mathbf{Z}_2$ symmetry is spontaneously broken (at least on a non-compact manifold) resulting in two light states in the limit of large $|M^2|$.

Let's see how this is repeated on the right-hand side. We can add a mass $\tilde{M}^2$ for the scalar. When $\tilde{M}^2 > 0$, we may integrate out the scalar, leaving ourselves with the trivial $\mathbf{Z}_2$ gauge theory $S_{\text{eff}} = i\pi \int t \cup S$. To count the states in the Hilbert space we can look at the partition function on a torus (i.e. $g = 1$). Using the normalisation (2.7), we have $Z = 2$ when $S = 0$ and $Z = 0$ otherwise, revealing again the existence of two light states. In contrast, when $\tilde{M}^2 < 0$, the $\mathbf{Z}_2$ gauge symmetry is broken. In this case, there is a unique ground state. The mapping is therefore

$$M^2 \quad \longleftrightarrow \quad -\tilde{M}^2,$$

which, in the statistical mechanics context, is the statement that Kramers-Wannier duality maps high temperatures to low temperatures.

We can also match the Hilbert spaces on the two sides, giving $\mathcal{B}_R = \tilde{\mathcal{B}}_R^+ \oplus \tilde{\mathcal{B}}_{NS}^+$ and $\mathcal{B}_{NS} = \tilde{\mathcal{B}}_R^- \oplus \tilde{\mathcal{B}}_{NS}^-$. Comparing these two expressions gives us the relation

$$\mathcal{B}_R^- = \tilde{\mathcal{B}}_{NS}^+, \tag{2.21}$$

where this is an equality about the spectrum of the Hamiltonian on $X = \mathbf{R} \times \mathbf{S}^1$ acting on these Hilbert spaces. This is the result previously advertised in (2.15).

**The Chiral Transformation is Kramers-Wannier Duality**

To end this section, we return to the question of how the discrete chiral transformation acts on our seed duality (2.9)

$$i\bar{\chi} \slashed{\mathcal{D}}_{S \cdot \rho} \chi \quad \longleftrightarrow \quad (\mathcal{D}_s \sigma)^2 + \sigma^4 + i\pi \Big[ \text{Arf}[s \cdot \rho] + \text{Arf}[\rho] + \int s \cup S \Big]. \tag{2.22}$$

As we have seen, the left-hand-side has an anomalous discrete chiral symmetry under which the action picks up an Arf invariant $\mathcal{I}[S \cdot \rho]$. The discussion above suggests that this should be realised as a Kramers-Wannier duality on the right-hand-side. It is simple to check that this is indeed the case. Performing a Kramers-Wannier duality gives

$$i\bar{\chi} \slashed{\mathcal{D}}_{S \cdot \rho} \chi \quad \longleftrightarrow \quad (\mathcal{D}_t \tilde{\sigma})^2 + \tilde{\sigma}^4 + i\pi \Big[ \text{Arf}[s \cdot \rho] + \text{Arf}[\rho] + \int s \cup (S + t) \Big]$$

$$\longleftrightarrow \quad (\mathcal{D}_t \tilde{\sigma})^2 + \tilde{\sigma}^4 + i\pi \, \text{Arf}[(S + t) \cdot \rho],$$

where the final expression arises from (2.8). Now, using (2.5), we have

$$i\bar{\chi}\slashed{D}_{S\cdot\rho}\chi \quad\longleftrightarrow\quad (\mathcal{D}_t\tilde{\sigma})^2 + \tilde{\sigma}^4 + i\pi\Big[\text{Arf}[t\cdot\rho] + \text{Arf}[\rho] + \int t\cup S + \text{Arf}[S\cdot\rho]\Big],$$

which coincides with our starting point (2.22), except for the extra anomalous $\text{Arf}[S\cdot\rho]$ term, matching the anomalous chiral transformation of the fermions.

# 3 Dirac Fermions and the XY-Model

In this section, we put together two copies of the "Majorana $\longleftrightarrow$ Ising" dualities to construct dualities which map Dirac fermions to variants of the XY-model. Among these is the original bosonization duality of Coleman [46], now dressed with appropriate spin structures and $\mathbf{Z}_2$ gauge fields. We will also see that Kramers-Wannier duality manifests itself as T-duality.

## 3.1 The Dirac Fermion

We work with a massless, complex, Dirac fermion $\psi$ with action

$$S_{\text{Dirac}} = \int_X i\bar{\psi}\slashed{D}_\rho\psi. \tag{3.23}$$

It will prove useful to review the various symmetries of this simple theory. In particular, there is a global symmetry

$$F = U(1)_L \times U(1)_R = \frac{U(1)_V \times U(1)_A}{\mathbf{Z}_2}, \tag{3.24}$$

with a well-known mixed 't Hooft anomaly between the two factors. There are also a number of discrete symmetries including charge conjugation which, with our choice of gamma matrices, is simply $\mathbf{Z}_2^C : \psi \mapsto \psi^\star$, which does not commute with $F$.

In what follows, we construct our Dirac fermion from two Majorana fermions

$$\psi = \chi_1 + i\chi_2.$$

The continuous chiral symmetries $F$ will not be manifest in many presentations below. Nonetheless, we will be able to track the action of $F$ using dualities and the action of a number of discrete $\mathbf{Z}_2$ symmetries. Specifically, we couple the Dirac fermion to two $\mathbf{Z}_2$ background gauge fields, $S$ and $C$ so that, written in the language of the previous section, the Dirac action (3.23) becomes

$$S_{\text{Dirac}} = \int i\bar{\chi}_1\slashed{D}_{S\cdot\rho}\chi_1 + i\bar{\chi}_2\slashed{D}_{(S+C)\cdot\rho}\chi_2. \tag{3.25}$$

Here the two global symmetries act as

$$\mathbf{Z}_2^S : \psi \mapsto -\psi \quad\text{and}\quad \mathbf{Z}_2^C : \psi \mapsto \psi^\star.$$

The first of these is part of the continuous symmetry group: $\mathbf{Z}_2^S \subset F$. Indeed, it is the element shared by both $U(1)_V$ and $U(1)_A$. The second factor $\mathbf{Z}_2^C$ is charge conjugation.

There are also two anomalous chiral transformations which (in Lorentzian signature) act as

$$\mathbf{Z}_2^{\mathrm{chi}} : \psi \mapsto \gamma^3 \psi \quad \text{and} \quad \mathbf{Z}_2^T : \begin{cases} \chi_1 \mapsto \chi_1 \\ \chi_2 \mapsto \gamma^3 \chi_2. \end{cases} \tag{3.26}$$

We'll see shortly why we refer to the second of these as $\mathbf{Z}_2^T$. (It is not time reversal!) Both have mixed anomalies with $\mathbf{Z}_2^S$ and $\mathbf{Z}_2^C$. From our discussion in Section 2, these can be written as

$$\mathbf{Z}_2^{\mathrm{chi}} : \qquad S_{\mathrm{Dirac}} \mapsto S_{\mathrm{Dirac}} + i\pi \left[ \mathrm{Arf}[C \cdot \rho] + \mathrm{Arf}[\rho] + \int S \cup C \right]$$

and

$$\mathbf{Z}_2^T : \qquad S_{\mathrm{Dirac}} \mapsto i\pi \mathrm{Arf}[(S+C) \cdot \rho].$$

We see that $\mathbf{Z}_2^{\mathrm{chi}}$ has a mixed anomaly with charge conjugation. In contrast, the second chiral transformation $\mathbf{Z}_2^T$ has a mixed anomaly with $\mathbf{Z}_2^S \subset F$. To better understand the role this plays, we write elements of the continuous symmetry groups as $g_{R/L} \in U(1)_{R/L}$ or, equivalently, as $g_{V/A} \in U(1)_{V/A}$ where

$$g_V^2 = g_L g_R \quad \text{and} \quad g_A^2 = g_L g_R^{-1}.$$

Then if $\eta \in \mathbf{Z}_2^T$, it is straightforward to check that $\eta g_L \eta = g_L$ and $\eta g_R \eta = g_R^{-1}$. In other words, conjugation by $\mathbf{Z}_2^T$ exchanges the vector and axial currents,

$$\eta g_V \eta = g_A \quad \text{and} \quad \eta g_A \eta = g_V.$$

The exchange of vector and axial symmetries is the defining feature of T-duality in $c = 1$ conformal field theories; we will return to this interpretation later.

## 3.2 Dirac = (Ising/$\mathbf{Z}_2$)$^2$

We now begin to explore the bosonization dualities for a Dirac fermion. We start with the free fermion (3.25)

$$S_{\mathrm{Dirac}} = \int i\bar{\chi}_1 \slashed{\partial}_{S \cdot \rho} \chi_1 + i\bar{\chi}_2 \slashed{\partial}_{(S+C) \cdot \rho} \chi_2. \tag{3.27}$$

From the previous section, this is dual to two copies of Ising/$\mathbf{Z}_2$,

$$S_{\mathrm{Dirac}} \quad \longleftrightarrow \quad \sum_{i=1}^{2} \left[ \int (\mathcal{D}_{s_i} \sigma_i^2) + \sigma_i^4 + i\pi \mathrm{Arf}[s_i \cdot \rho] \right] + i\pi \left[ \int S \cup (s_1 + s_2) + s_2 \cup C \right]. \tag{3.28}$$

The continuous symmetry $F$ is not manifest in the UV scalar Lagrangian; in particular, the $\sigma_i^4$ terms mean that there is no $U(1)$ symmetry acting on $\sigma_1 + i\sigma_2$. At first glance, only the $\mathbf{Z}_2^S \subset U(1)_V$ symmetry is visible in the Ising theory. We can do a little better than this. In fact, there is another manifest $\mathbf{Z}_2$ action present in (3.28), namely $\sigma_1 \leftrightarrow \sigma_2$, under which we must simultaneously swap $S \leftrightarrow S + C$. The corresponding symmetry of the fermionic theory exchanges $\chi_1 \leftrightarrow \chi_2$, which is a combination of charge conjugation and a $U(1)_V$ rotation by $\pi/2$. This furnishes the Ising theory with a manifest $\mathbf{Z}_4 \rtimes \mathbf{Z}_2^C = D_4$ symmetry.

Nonetheless, the duality tells us that the full continuous symmetry must be present. Referring to the operator map (2.16), we see that schematically the $U(1)_V$ symmetry acts on

$\phi_V(x) = \mu_1(x)\sigma_1(x) + i\mu_2(x)\sigma_2(x)$, where $\mu_i$ are the disorder operators which carry $\mathbf{Z}_2$ gauge charge, and must therefore be dressed by $\sigma_i$ excitations.

To construct the operator that transforms under $U(1)_A$, it is simplest to perform a T-duality $\mathbf{Z}_2^T$ which, as we have seen, transposes $U(1)_V$ and $U(1)_A$. This is implemented by a chiral transformation of $\chi_2$ on the fermionic side, which maps to a Kramers-Wannier duality in the bosonic language. We therefore replace $\sigma_2$ with the dual field, $\tilde{\sigma}_2$. With the now-familiar manipulations, using (2.5) and (2.8), we find the dual theory can be written as

$$S_{\text{dual}} = \int (\mathcal{D}_{s_1}\sigma_1)^2 + \sigma_1^4 + (\mathcal{D}_{s_2}\tilde{\sigma}_2)^2 + \tilde{\sigma}_2^4 + i\pi\left[\sum_{i=1}^{2}\left[\text{Arf}[s_i \cdot \rho] + \int s_i \cup S\right]\right.$$
$$\left. + \int s_2 \cup C + \text{Arf}[(S+C)\cdot\rho]\right]. \qquad (3.29)$$

This theory is very almost self-dual; the new theory coincides with (3.28), apart from the final term involving background fields. This, of course, is inherited from the behaviour of the Ising/$\mathbf{Z}_2$ theory discussed in the previous section. There is now a natural action of $U(1)_A$ on the local operator $\phi_A(x) = \mu_1(x)\sigma_1(x) + i\tilde{\mu}_2(x)\tilde{\sigma}_2(x)$.

The self-duality of (3.28) is not what we would usually call T-duality. Moreover, it is surprising that the free fermion corresponds to the self-dual point. We will return to this in Section 3.5. But first, we turn to a different model where we will make contact with the more familiar description of T-duality.

## 3.3 Dirac/Z$_2$ = XY-Model

We can derive a duality for Dirac fermion coupled to a $\mathbf{Z}_2$ gauge field starting from the duality (3.28). To this end, we first add $\text{Arf}[S \cdot \rho]$ and subsequently promote $S$ to a dynamical gauge field. After some manipulations, this results in the following duality:

$$\int i\bar{\chi}_1 \slashed{D}_{(s+S)\cdot\rho}\chi_1 + i\bar{\chi}_2 \slashed{D}_{(s+S+C)\cdot\rho}\chi_2 + i\pi\text{Arf}[s\cdot\rho]$$
$$\longleftrightarrow \quad \sum_{i=1}^{2}\left[\int (\mathcal{D}_{s_i}\sigma_i)^2 + \sigma_i^4 + i\pi\int S\cup s_i\right] + i\pi\left[\int s_1\cup s_2 + \int s_2\cup C\right]. \quad (3.30)$$

The continuous chiral symmetry of the fermionic theory is $F' = F/\mathbf{Z}_2$, with $F$ defined in (3.24). The fact that we have gauged $\mathbf{Z}_2 = (-1)^F$ means that the Lorentz group $SO(1,1)$ acts faithfully on these theories, rather than $\text{Spin}(1,1)$. Relatedly, the fermionic theory is independent of the fiducial spin structure $\rho$. The $\mathbf{Z}_2$ symmetry which exchanges $\chi_1 \longleftrightarrow \chi_2$ and $S \longleftrightarrow S+C$ remains. It is not hard to see that the bosonic theory also still enjoys this symmetry with $\sigma_1 \longleftrightarrow \sigma_2$.

Once again, the action of the continuous chiral symmetry $F'$ is hidden in the scalar theory. It must act on kink states. The $\int s_1 \cup s_2$ term plays an important role, endowing the disorder operator $\mu_1$ with $s_2$ gauge charge, and vice-versa. This means that the $U(1)_V$ vector symmetry acts on an operator which schematically takes the form

$$\phi_V \sim \mu_1(x)\sigma_2(x) + i\mu_2(x)\sigma_1(x).$$

Now T-duality acts less trivially on the scalar theory. We perform a Kramers-Wannier duality on $\sigma_2$, to find a dual description with just a single dynamical $\mathbf{Z}_2$ gauge field,

$$S_{\text{dual}} = \int (\mathcal{D}_s\sigma_1)^2 + \sigma_1^4 + (\mathcal{D}_{s+S+C}\tilde{\sigma}_2)^2 + \tilde{\sigma}_2^4 + i\pi\int s\cup S. \qquad (3.31)$$

This formulation masks the $S \longleftrightarrow S + C$ symmetry. The fact that the quantum theory does exhibit this symmetry is reminiscent of the conjectured self-duality seen of QED in 3d, coupled to a pair of bosons [24] or a pair of fermions [25]. This self-duality, which holds only under the assumption that these theories flow to an IR fixed point, can also be seen through the 3d duality web [1, 28].

We can partially explore the phase structure of (3.30) by adding the unique $D_4$ invariant mass term which, in Lorentzian signature, is $\bar{\psi}\psi \longleftrightarrow -(\sigma_1^2 + \sigma_2^2)$. (The mass term $\bar{\psi}\gamma^3\psi$ is not invariant under $\mathbf{Z}_2^C \subset D_4$, while the mass term $\sigma_1^2 - \sigma_2^2$ is not invariant under the $\mathbf{Z}_2 \subset D_4$ which exchanges $\sigma_1$ and $\sigma_2$.) It is simple to see that both sides of the duality (3.30) exhibit a trivial phase for one sign of the mass, and a topological phase with

$$S_{\text{eff}} = i\pi \int S \cup C \tag{3.32}$$

for the other sign.

**The Compact Boson Description**

The equivalence of the bosonic theories (3.30) and (3.31) is not usually what comes to mind when we think of T-duality. Nonetheless, we claim that these are equivalent. The purpose of this section is to make contact with the more familiar language.

Two Ising models, coupled to some combination of $\mathbf{Z}_2$ gauge theories, form a $c = 1$ CFT. If this CFT is independent of the background spin structure (i.e. like (3.30) rather than (3.28)) then it can be described by one of two objects: a compact boson, or an orbifold. (For good reason, the canonical review for this subject remains [61].) The scalar theory (3.30) falls into the former class: it can be described in terms of a compact scalar field $\theta \in [0, 2\pi]$. Roughly speaking, this should be viewed as the phase of the complex field $\phi_V$. If we ignore, for now, the coupling to the background $C$ field then the action takes the form

$$S_\theta = \int \frac{R^2}{8\pi} (\mathcal{D}_S \theta)^2. \tag{3.33}$$

It is difficult to determine $R^2$, the radius of the boson, directly from (3.30). Matching correlators of chiral bosons with chiral fermions gives the well-known answer $R = 2$. We will see below that we can, in fact, determine this radius using duality arguments alone.

The background gauge field $S$ in (3.33) is associated to the symmetry

$$\mathbf{Z}_2^S : \theta \mapsto \theta + \pi.$$

Meanwhile, the charge conjugation symmetry coupled to the background field $C$ acts as

$$\mathbf{Z}_2^C : \theta \mapsto -\theta.$$

The discrete chiral transformation $\mathbf{Z}_2^T$ is identified with T-duality, which now acts in the familiar fashion. The only slight subtlety is the existence of the background $\mathbf{Z}_2$ gauge field $S$. This, however, is easily dealt with by extending the theory to a include a background gauge field $V$ for $U(1)_V$ which acts as $\theta \mapsto \theta + \alpha$. Since $\mathbf{Z}_2^S \subset U(1)_V$, we can always subsequently restrict to the $\mathbf{Z}_2$ subgroup. The standard T-duality transformation now maps

$$\int \frac{R^2}{8\pi} (d\theta - V)^2 + \frac{i}{2\pi} d\theta \wedge d\tilde{\theta} \quad \longleftrightarrow \quad \int \frac{1}{2\pi R^2} (d\tilde{\theta})^2 + \frac{i}{2\pi} \tilde{\theta} dV,$$

where $\tilde{\theta} \in [0, 2\pi)$. We recover the standard result that the T-dual scalar $\tilde{\theta}$ acts as a theta angle for the background gauge field $V$, with T-duality mapping

$$\text{T-duality}: \ R \mapsto \frac{2}{R}. \tag{3.34}$$

Now restrict $V$ to a $\mathbf{Z}_2$ gauge field; the resulting coupling measures the winding $\frac{1}{2\pi}\int d\tilde{\theta} \in H^1(X; \mathbf{Z})$ mod 2. We write the resulting coupling as

$$S_{\tilde{\theta}} = \int \frac{1}{2\pi R^2}(d\tilde{\theta})^2 + \frac{i}{2}\int d\tilde{\theta} \cup S. \tag{3.35}$$

This is equivalent to the description (3.31).

As we mentioned above, it is difficult to fix the radius of the boson $R^2$ directly from (3.30) as this is a marginal parameter. Nonetheless, there is a rather slick way to determine the value. To see this, first promote the background field $S$ in (3.30) to become dynamical. After renaming various fields, this results in the Ising-type theory

$$S_{\text{new}} = \int (\mathcal{D}_s \sigma_1)^2 + \sigma_1^4 + (\mathcal{D}_{s+S}\tilde{\sigma}_2)^2 + \tilde{\sigma}_2^4 + i\pi \left[ \int s \cup (S + C) + \int S \cup C \right].$$

Up to a relabelling of background fields, this coincides with the theory (3.31) that arises from T-duality. We learn that gauging $\mathbf{Z}_2^S$ gives another path to reach the T-dual description. We can, of course, do this starting from the compact boson (3.33). Gauging $\mathbf{Z}_2^S : \theta \mapsto \theta + \pi$ simply restricts the range to $\theta \in [0, \pi)$. Defining a new variable $\hat{\theta} = 2\theta \in [0, 2\pi)$, we have

$$S_{\text{new}} = \int \frac{R^2}{4 \cdot 8\pi}(d\hat{\theta})^2. \tag{3.36}$$

But we have seen that this should coincide with the T-dual description (3.35). In other words,

$$\frac{R^2}{32\pi} = \frac{1}{2\pi R^2} \quad \Rightarrow \quad R = 2,$$

which is the expected answer for a compact boson dual to a free fermion!

The phase structure of the theories (3.30) implies an interesting anomaly for the compact boson. By the usual arguments [46], the mass term $\bar{\psi}\psi$ is equivalent to $-\cos\tilde{\theta}$. Note, in particular, that this even under both $\mathbf{Z}_2^S$ and $\mathbf{Z}_2^C$. However, a change of sign of the mass term can be effected by the shift $\tilde{\theta} \to \tilde{\theta} + \pi$. The result (3.32) tells us that under such a shift, the partition function of the compact boson must have an anomalous shift by $i\pi \int S \cup C$. This is reminiscent of the famous axial anomaly, under which a shift of $\tilde{\theta}$ is anomalous in the presence of a background field for $U(1)_V$. Indeed, $\mathbf{Z}_2^S \subset U(1)_V$. However, the anomaly here is different and depends, crucially, on the background field for $\mathbf{Z}_2^C$. It is closely related to the $\mathbf{Z}_2 \times \mathbf{Z}_2$ anomaly described in [62].

**The Usual Bosonization Dictionary**

The fact that a compact boson is not equivalent to a Dirac fermion, but rather to a Dirac fermion coupled to a $\mathbf{Z}_2$ gauge field, is seen most simply by comparing the partition functions. These agree only when the fermion is summed over all boundary conditions [60], a procedure that is equivalent to coupling to a $\mathbf{Z}_2$ gauge field. Although this has been known for many years, the need to include a $\mathbf{Z}_2$ quotient is a point that is omitted in most textbook discussions of

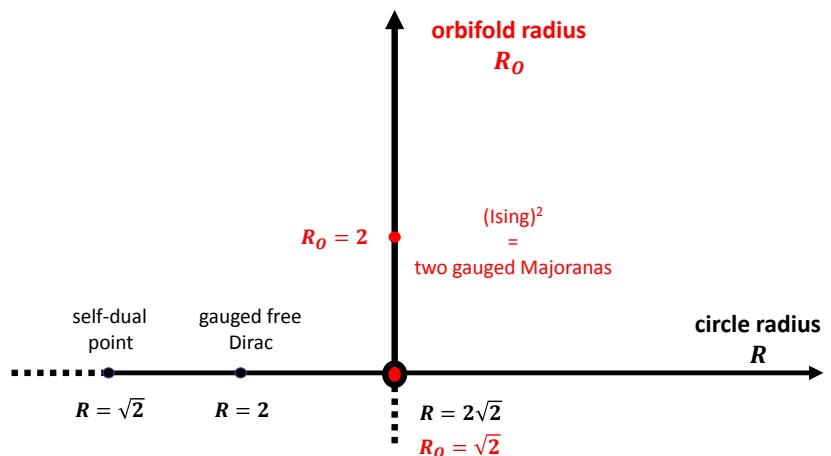

Figure 1: The moduli space of bosonic $c = 1$ CFTs with marginal deformations.

bosonization. For this reason, we briefly review how it arises in the standard bosonization story that we learn in school.

We decompose the compact boson into its left- and right-moving constituents as

$$\theta = \frac{1}{2}(\theta_L - \theta_R) \quad \text{and} \quad \tilde{\theta} = \theta_L + \theta_R.$$

One can then show that, for the special choice of $R = 2$, the correlation functions (or, equivalently, OPEs) of free, chiral fermions $\psi_L$ and $\psi_R$ are reproduced if we make the identification

$$\psi_L = \sqrt{\frac{1}{2\pi\epsilon}} e^{-i\theta_L} \quad \text{and} \quad \psi_R = \sqrt{\frac{1}{2\pi\epsilon}} e^{+i\theta_R}, \tag{3.37}$$

with $\epsilon$ a UV cut-off with dimensions of length. However, neither of these operators exist as local operators in the bosonic theory – as indeed, they cannot in the absence of a choice of spin structure. The well-defined operators in the theory of a compact boson are $e^{in\theta + im\tilde{\theta}}$ with $n, m \in \mathbf{Z}$. This means, for example, that $e^{i\tilde{\theta} - 2i\theta} = e^{2i\theta_R}$ is allowed, but $e^{i\theta_R}$ is not. On the other hand, $e^{i\theta} = e^{\frac{1}{2}\theta_L - \frac{1}{2}\theta_R}$ is permitted. In this way, a compact boson (of the correct radius) is equivalent to a free fermion such that the states with $(-1)^F = -1$ are projected out, but certain additional states included. This is precisely the Dirac fermion coupled to a $\mathbf{Z}_2$ gauge field, with operators like $e^{i\theta}$ reflecting the existence of the twisted sector.

There is one final comment that completes the circle of ideas in this section. We have identified T-duality as Kramers-Wannier duality in the Ising description (3.30) which, in turn, maps to the discrete chiral transformation $\mathbf{Z}_2^T$, defined in (3.26), on the fermion. This discrete chiral symmetry leaves $\psi_L$ alone and transforms $\psi_R \to \psi_R^\dagger$. But this, in turn, acts as $\mathbf{Z}_2^T : \theta_R \mapsto -\theta_R$, which is the standard action of T-duality.

### 3.4 Bosonic $c = 1$ CFTs

The landscape of bosonic $c = 1$ conformal field theories is shown in Figure 1. (A pedagogical discussion of these theories and their moduli space can be found in [61]. We are omitting

isolated theories.) There are two branches, corresponding to the compact boson with target space $\mathbf{S}^1$, shown on the horizontal axis, and the orbifold, with target space $\mathbf{S}^1/\mathbf{Z}_2$, shown on the vertical axis. The two axes meet at the Kosterlitz-Thouless point.

In the compact boson description, the marginal parameter is the radius $R$. In the fermionic description, the marginal operator is provided by the Thirring coupling. In Euclidean signature, this corresponds to

$$\Delta S = \frac{g}{2} \int_X (\bar{\psi}\gamma^\mu \psi)(\bar{\psi}\gamma_\mu \psi).$$

The dictionary between this and the bosonic parameter was first derived by Coleman [46]. The radius $R$ of the compact boson $\theta$, which shifts under $U(1)_V$, is related to the Thirring coupling by

$$\left(\frac{2}{R}\right)^2 = 1 + \frac{g}{\pi}. \tag{3.38}$$

As we have seen previously, the Ising dual (3.30) corresponds to $R = 2$. It is natural to ask: can one construct Ising descriptions of other points in the moduli space?

The Thirring coupling maps to $\sigma_1^2 \sigma_2^2$ in the Ising language. (Alternatively, we could use disorder operators $\mu_1^2 \mu_2^2$; both include the marginal operator in the OPE.) However, identifying the exact operator map in the UV, to give a dictionary analogous to (3.38), is more tricky. Nonetheless, there are a number of manipulations that we can do to construct Ising-like descriptions of special points in the moduli space of $c = 1$ CFTs.

**The Self-Dual Point**

When the compact boson (3.33) sits at the self-dual radius $R = \sqrt{2}$, the theory is known to exhibit a symmetry enhancement, with the $F = U(1)_L \times U(1)_R$ current algebra enlarged to $SU(2)_L \times SU(2)_R$. Here we present a derivation of this symmetry enhancement using the duality web.

To construct the self-dual point, we start with the theory (3.30) of a Dirac/$\mathbf{Z}_2$ fermion. We write the dual scalar as an $R = 2$ compact boson (3.33). Both sides have an explicit $U(1)_V$ global symmetry, and we may couple this to a background $U(1)$ gauge field, resulting in the duality

$$S_{\text{Dirac}}[A; s] \quad \longleftrightarrow \quad S_{R=2}[A],$$

where

$$S_{\text{Dirac}}[A; s] = \int i\bar{\psi}\slashed{D}_{s+A}\psi + i\pi \operatorname{Arf}[s \cdot \rho]$$

and

$$S_{R=2}[A] = \int \frac{1}{2\pi}(\partial\theta - A)^2.$$

Now take two such dualities and gauge the diagonal $\mathbf{Z}_2$ symmetry. We get the new duality,

$$S_{\text{Dirac}}[A_1 + u; s] + S_{\text{Dirac}}[A_2 + u; t] \quad \longleftrightarrow \quad S_{R=2}[A_1 + u] + S_{R=2}[A_2 + u].$$

Defining $v = u + s$ and $w = s + t$, we may combine the Arf terms in $S_{\text{Dirac}}$ to give

$$S_{\text{Dirac}}[A_1 + v + s; s] + S_{\text{Dirac}}[A_2 + v + w + t; t] = \int i\bar{\psi}_1 \slashed{D}_{A_1 + v}\psi_1 + i\bar{\psi}_2 \slashed{D}_{A_2 + v + w}\psi_2$$

$$+ i\pi\left[\text{Arf}[w \cdot \rho] + \text{Arf}[\rho] + \int w \cup t\right].$$

But the $\mathbf{Z}_2$ gauge field $t$ now acts only to set $w = 0$. We're left with two Dirac fermions coupled to the dynamical gauge field $v$,

$$\int i\bar{\psi}_1 \slashed{D}_{v+A_1}\psi_1 + i\bar{\psi}_2 \slashed{D}_{v+A_2}\psi_2 \quad \longleftrightarrow \quad S_{R=2}[A_1 + u] + S_{R=2}[A_2 + u]. \tag{3.39}$$

In this formulation, the fermionic theory has a manifest $U(2)/\mathbf{Z}_2$ global symmetry. It is convenient to write

$$A_\pm = A_1 \pm A_2,$$

as then $A_\pm$ are correctly normalized $U(1)$ fields, since the $\mathbf{Z}_2$ gauge field $v$ imposes that all physical operators carry charge of the same parity under $A_1$ and $A_2$. We see find that $A_+$ couples to an overall $U(1)$, whilst $A_-$ couples to a $U(1) \subset SU(2)$.

From the above dualities, we have equivalent descriptions of this theory in terms of bosonic variables. Importantly, the dynamical $\mathbf{Z}_2$ gauge field $u$ ensures that the two combinations $\theta_\pm = \theta_1 \pm \theta_2$ are well-defined $2\pi$ periodic variables. Inverting this relation, we find that the bosonic kinetic terms are shifted by a factor of two,

$$S_{R=2}[u + A_1] + S_{R=2}[u + A_2] \quad \longleftrightarrow \quad \int \frac{1}{4\pi}(\partial\theta_+ - A_+)^2 + \frac{1}{4\pi}(\partial\theta_- - A_-)^2, \tag{3.40}$$

so that we are left with two decoupled scalars, each with the radius $R = \sqrt{2}$. It is also straightforward to verify that there is a third description of this theory in terms of four Ising fields, most symmetrically written as follows:

$$\int i\bar{\psi}_1 \slashed{D}_v \psi_1 + i\bar{\psi}_2 \slashed{D}_v \psi_2 \quad \longleftrightarrow \quad \int \frac{1}{4\pi}(\partial\theta_+)^2 + \frac{1}{4\pi}(\partial\theta_-)^2$$

$$\longleftrightarrow \quad \sum_{i=1}^{4}\left[\int (\mathcal{D}_{s_i}\sigma_i)^2 + \sigma_i^4 + i\pi\int v \cup s_i\right] + i\pi\int \sum_{i<j} s_i \cup s_j.$$

Neither of the scalar theories exhibits a non-Abelian global symmetry but must, nonetheless, inherit one from the duality. We now promote $A_+$ to a dynamical $U(1)$ gauge field. On the scalar side, this simply kills $\theta_+$, leaving us with a single compact scalar. On the fermionic side, it is convenient to absorb $v$ into the new dynamical gauge field $a$ (which should be taken to be a $\text{Spin}_C$ field), leaving a $U(1)$ gauge theory containing two fermions which exhibits the non-Abelian global symmetry,

$$\int i\bar{\psi}_1 \slashed{D}_{a+A_-/2}\psi_1 + i\bar{\psi}_2 \slashed{D}_{a-A_-/2}\psi_2 \quad \longleftrightarrow \quad \int \frac{1}{4\pi}(\partial\theta_- - A_-)^2. \tag{3.41}$$

We recover the well-known result that a compact scalar with radius $R^2 = 2$ exhibits an enhanced non-Abelian global symmetry.

One of the special properties of this point is that it has a symmetry $S \leftrightarrow C$. Concretely, $\mathbf{Z}_2^S \subset U(1)_{A_-}$ is embedded in the $SU(2)$ symmetry on the left-hand side, and we claim that

$\mathbf{Z}_2^C$ is also an $SU(2)$ rotation. To see this, note that $\theta_- \to -\theta_-$ is equivalent to $\theta_1 \leftrightarrow \theta_2$ and therefore to $\psi_1 \leftrightarrow \psi_2$. This is an element of $SU(2)$ up to a gauge transformation. Since all $\mathbf{Z}_2$ elements of $SU(2)$ are conjugate to each other, it now follows that $S \leftrightarrow C$ is a symmetry.[4]

**(Majorana/$\mathbf{Z}_2$)$^2$ = Orbifold**

We turn now to the orbifold CFT with target space $\mathbf{S}^1/\mathbf{Z}_2$. It is straightforward to give description of the orbifold theory at the free fermion point. Here we choose to approach this by gauging the the charge conjugation symmetry $C$ in the Dirac = XY-model duality of (3.30). After relabelling various fields, the fermionic theory becomes

$$\sum_{i=1}^2 \left[ \int i\bar{\chi}_i \slashed{D}_{s_i \cdot \rho} \chi_i + i\pi \int s_i \cup C \right] + i\pi \left[ \mathrm{Arf}[s_1 \cdot \rho] + \mathrm{Arf}[S \cdot \rho] + \mathrm{Arf}[\rho] + \int s_1 \cup S \right].$$

This is two copies of the Majorana/$\mathbf{Z}_2$ theory, one in the form (2.17) and the other in the form (2.19). Meanwhile, the scalar theory from (3.30) gives the dual description

$$\longleftrightarrow \quad \int (\mathcal{D}_s \sigma_1)^2 + \sigma_1^4 + \int (\mathcal{D}_C \sigma_2)^2 + \sigma_2^4 + i\pi \left[ \int s \cup (S+C) + \int S \cup C \right].$$

Alternatively, we could perform a chiral transformation on $\chi_1$ and, correspondingly, a Kramers-Wannier duality to $\sigma_1$ to get the duality in the form

$$\sum_{i=1}^2 \left[ \int i\bar{\chi}_i \slashed{D}_{s_i \cdot \rho} \chi_i + i\pi \int s_i \cup C \right] + i\pi \left[ \mathrm{Arf}[S \cdot \rho] + \mathrm{Arf}[\rho] + \int s_1 \cup S \right]$$

$$\longleftrightarrow \quad \int (\mathcal{D}_{S+C} \tilde{\sigma}_1)^2 + \tilde{\sigma}_1^4 + \int (\mathcal{D}_C \sigma_2)^2 + \sigma_2^4 + i\pi \int S \cup C, \tag{3.42}$$

which makes manifest the usual (Majorana/$\mathbf{Z}_2$)$^2$ = (Ising)$^2$ form of this duality.

In either case, the action on the compact boson is $\mathbf{Z}_2^C : \theta \mapsto -\theta$. Conventionally, the resulting orbifold is taken to have "radius" $R_O = 2$.

We can again vary the marginal parameter to move along the line of orbifold CFTs. As previously, this is effected by the UV coupling $\sigma_1^2 \sigma_2^2 \sim -\tilde{\sigma}_1^2 \tilde{\sigma}_2^2$. A generic value of this coupling requires us to tune the mass terms in order to hit a point on the orbifold line. The story is a little different when we add the coupling to the point where there is an enhanced $U(1)$ symmetry. In the formulation (3.42), this occurs with

$$S_{XY} = \int (\mathcal{D}_{S+C} \tilde{\sigma}_1)^2 + (\mathcal{D}_C \sigma_2)^2 + m^2(\tilde{\sigma}_1^2 + \sigma_2^2) + \lambda(\tilde{\sigma}_1^2 + \sigma_2^2)^2 + i\pi \int S \cup C.$$

This, of course, is the usual description of the XY-model. Varying $m^2$ now allows us to access the line of $c = 1$ theories with $\mathbf{S}^1$ target space. This intersects the orbifold line at the Kosterlitz-Thouless point.

This raises an interesting question. What is the analogous "$U(1)$ limit" of the theory (3.31), where the Ising scalars are also coupled to a dynamical $\mathbf{Z}_2$ gauge field? This now also gives us the XY-model, but where the phase of the complex scalar $\sigma_1 + i\tilde{\sigma}_2$ is acted upon by $\mathbf{Z}_2^S$. This results in another branch of the XY-model with $\mathbf{S}^1$ target space. This meets the original branch at the self-dual point: indeed, the two branches of $c = 1$ CFTs are related by an $SU(2)$ transformation.

---

[4]This statement is presented in the language of partition functions as equation (B.57) in Appendix B.

One upshot of this is that the Kosterlitz-Thouless point and the self-dual point must be related by gauging charge conjugation $C$. This follows straightforwardly from the $S \leftrightarrow C$ symmetry we highlighted above [61].

### 3.5 Fermionic $c = 1$ CFTs

In the previous section, we described the landscape of modular invariant $c = 1$ CFTs. These do not require a choice of spin structure. Our purpose in this section is to describe the analogous moduli space for $c = 1$ theories which are sensitive to the choice of spin structure. In Figure 2, we show the space of such theories that are continuously connected to free fermion points. Our purpose in this section (and in Appendix B) is to flesh out the structure of this diagram.

**The Dirac Fermion Revisited**

We already discussed the Dirac fermion in Section 3.2, where we offered a dual Ising description in (3.28). However, one might wonder if there is a simpler dual description in terms of a compact boson.

A detailed answer to this question was provided long ago in [63, 64]. Here we offer a slightly different take which connects to our larger duality web. We can start with the duality (3.30). If we add $\text{Arf}[S \cdot \rho] + \text{Arf}[\rho]$, and subsequently promote $S$ to a dynamical gauge field, then we get back to the original Dirac fermion (3.27) and the duality (3.28). We can, however, perform the same steps on the compact boson (3.33). This gives the dual of a Dirac fermion to be

$$S_{\text{Dirac}} \quad \longleftrightarrow \quad \int \frac{1}{2\pi}(D_{s+S}\theta)^2 + i\pi\text{Arf}[s \cdot \rho]. \tag{3.43}$$

(We continue to suppress $C$ on the right-hand side; as before, it couples to the $\theta \mapsto -\theta$ symmetry.) The presence of the Arf invariant $\text{Arf}[s \cdot \rho]$ is important. First, it means that the dynamical $\mathbf{Z}_2$ gauge field does not simply halve the period of $\theta$. Secondly, it ensures that this theory depends on the fiducial spin structure, as it must if it is to be dual to a Dirac fermion.

We can also provide a T-dual description. In fact, it will prove useful to consider a compact boson $\theta$ of arbitrary radius $R$. The usual T-duality (3.35) gives

$$\int \frac{R^2}{8\pi}(D_s\theta)^2 + i\pi\text{Arf}[s \cdot \rho] \quad \longleftrightarrow \quad \int \frac{(2/R)^2}{8\pi}(\partial\tilde{\theta})^2 + i\pi\left[\frac{1}{2\pi}\int d\tilde{\theta} \cup s + \text{Arf}[s \cdot \rho]\right].$$

We now define $\hat{\theta} = \tilde{\theta}/2$. Clearly $\hat{\theta} \in [0, \pi)$. To implement this, we take $\hat{\theta} \in [0, 2\pi)$ but introduce a new $\mathbf{Z}_2$ gauge field $t$ which gauges the shift symmetry $\hat{\theta} \to \hat{\theta} + \pi$. Now consider the winding $n = \frac{1}{2\pi}\int d\tilde{\theta}$ around some cycle. If the winding is $n$ odd, then $\frac{1}{2\pi}\int d\hat{\theta}$ is half-integer; this is allowed only when we include a non-vanishing holonomy for $t$. The upshot is that the T-dual theory can be equivalently written as

$$\frac{R^2}{8\pi}(D_s\theta)^2 + i\pi\text{Arf}[s \cdot \rho] \quad \longleftrightarrow \quad \int \frac{(4/R)^2}{8\pi}(D_t\hat{\theta})^2 + i\pi\left[\int t \cup s + \text{Arf}[s \cdot \rho]\right]$$

$$\longleftrightarrow \quad \int \frac{(4/R)^2}{8\pi}(D_t\hat{\theta})^2 + i\pi\text{Arf}[t \cdot \rho] + i\pi\text{Arf}[\rho],$$

where, in the second step, we have used (2.11). Up to the background $\text{Arf}[\rho]$ term, we see that T-duality for this compact boson acts as

$$\text{T-duality}: R \mapsto \frac{4}{R}. \tag{3.44}$$

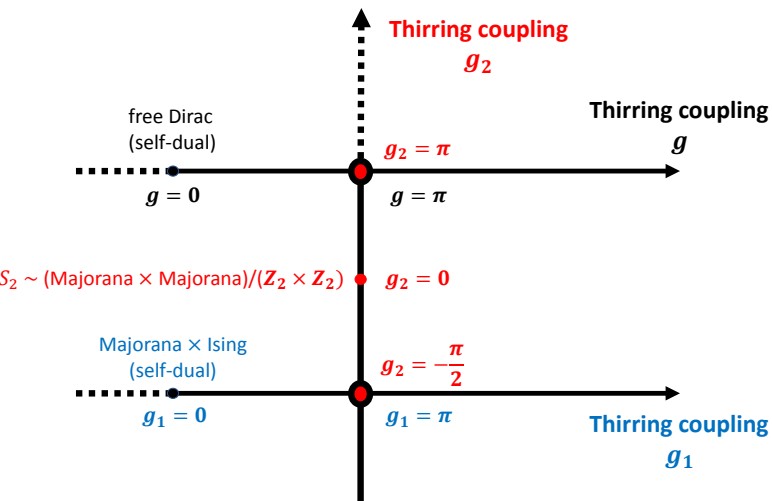

Figure 2: The moduli space of fermionic $c = 1$ CFTs connected to the free Dirac point.

This is to be contrasted with the more familiar result (3.34). Of course, there is little mystery here: as the derivation above shows, the extra factor of 2 comes because the Dirac fermion is dual to a compact boson coupled to a $\mathbf{Z}_2$ gauge field. In Appendix B, we show that the partition function of the appropriate compact boson is indeed invariant under this map.

Nonetheless, this does explain an observation that we met in Section 3.2. The free Dirac fermion – corresponding to $R = 2$ – sits at the self-dual point (up to an anomaly involving background fields). This is distinct from the bosonic CFT of a Dirac/$\mathbf{Z}_2$ fermion where the self-dual point has enhanced $SU(2)$ symmetry and arises only after adding a marginal deformation.

We can now ask what happens as we deform away from the Dirac point. The Thirring coupling again corresponds to the radius of the circle through Coleman's operator map (3.38). The resulting set of theories is shown in the upper horizontal line in Figure 2. The T-duality map (3.44) can then be interpreted as a map on the Thirring coupling[5]

$$\text{T-duality}: \ g \mapsto -\frac{g}{1 + g/\pi}. \tag{3.45}$$

Once again, we see that the free fermion $g = 0$ is the self-dual point. For small $g$, we have $g \mapsto -g$, reflecting the fact that T-duality is a chiral transformation under which the Thirring term is odd.

**Orbifold CFTs with a Spin Structure**

There are two further free fermion theories that we can construct, both of which depend on the background spin structure $\rho$. Each of these arises by gauging a suitable $\mathbf{Z}_2$ symmetry of the free Dirac fermion

$$S_{\text{Dirac}}[S, C] = \int \ i\bar{\chi}_1 \slashed{\partial}_{S \cdot \rho} \chi_1 + i\bar{\chi}_2 \slashed{\partial}_{(S+C) \cdot \rho} \chi_2. \tag{3.46}$$

---

[5]For the Dirac/$\mathbf{Z}_2$ fermion, the corresponding T-duality map (3.34) gives

$$\text{T-duality}: \ g \mapsto \frac{3\pi - g}{1 + g/\pi}.$$

Now the self-dual point sits at $g = \pi$. The free theory at $g = 0$ is mapped to $g = 3\pi$.

As we saw in Section 3.3, gauging $\mathbf{Z}_2^S$ gives the Dirac/$\mathbf{Z}_2$ theory which is independent of the spin structure. However, we have other options:

- Majorana × (Majorana/$\mathbf{Z}_2$) = Majorana × Ising: If we gauge the $\mathbf{Z}_2^C$ symmetry of the Dirac fermion, we have the free fermion theory:

$$S_1[S,C] = \int i\bar{\chi}_1 \slashed{D}_{S\cdot\rho}\, \chi_1 + i\bar{\chi}_2 \slashed{D}_{(S+t)\cdot\rho}\, \chi_2 + i\pi \int t \cup C. \tag{3.47}$$

  This is depicted on the lower horizontal line in Figure 2.

- (Majorana × Majorana)/($\mathbf{Z}_2 \times \mathbf{Z}_2$): If we gauge both $\mathbf{Z}_2^S$ and $\mathbf{Z}_2^C$ and couple them to appropriate background fields, we arrive at a second free fermion theory:

$$S_2[S,C] = \int i\bar{\chi}_1 \slashed{D}_{(s+S)\cdot\rho}\, \chi_1 + i\bar{\chi}_2 \slashed{D}_{(s+t+S+C)\cdot\rho}\, \chi_2 + i\pi\mathrm{Arf}[s\cdot\rho] + i\pi\mathrm{Arf}[t\cdot\rho]. \tag{3.48}$$

  This is depicted on the vertical line in Figure 2.

Neither $S_1$ nor $S_2$ have continuous $U(1)$ symmetries. In this sense, they are fermionic versions of the familiar orbifold CFT with target space $\mathbf{S}_1/\mathbf{Z}_2$.

Each of the theories (3.46), (3.47) and (3.48) has a marginal Thirring deformation. We call the associated couplings $g$, $g_1$ and $g_2$ respectively. The question that we would like to ask is: do the marginal lines of these three theories intersect? We claim that the answer is yes, with the result shown in Figure 2.

There are two ways to answer this question. It is possible to use the duality web developed in this paper and, in particular, deploy arguments similar to those used in Section 3.4. As discussed in section 3.4, the self-dual theory (3.41) has a symmetry $S \leftrightarrow C$. This, it turns out, is sufficient to derive the intersection points shown in Figure 2.

However, at this point we feel it is appropriate to resort to more traditional language. To this end, we introduce the torus partition function for a compact boson of radius $R$,

$$Z_\theta(S,C;R) = \int \mathcal{D}\theta \exp\left[-\int \frac{R^2}{8\pi}(D_{S,C}\theta)^2\right].$$

Note that, in contrast to previous equations like (3.33), we have explicitly included background gauge fields for both $\mathbf{Z}_2^S : \theta \to \theta + \pi$ and $\mathbf{Z}_2^C : \theta \to -\theta$. This partition function is well known, and a number of its properties are collected in Appendix B.

Each of the three theories (3.46), (3.47) and (3.48) has a straightforward description in terms of the compact boson, allowing us to straightforwardly compute their partition functions. These depend on the radius $R$ which, in each case, maps to the Thirring coupling of the fermion through Coleman's result (3.38):

$$\frac{4}{R^2} = 1 + \frac{g}{\pi}. \tag{3.49}$$

In particular, the free fermion point corresponds to $R = 2$.

### Dirac

The duality between the Dirac fermion and a compact boson was given in (3.43). Using this, we can write the partition function of the Dirac fermion on a torus as

$$Z_{\mathrm{Dirac}}(S,C;\rho;R) = \frac{1}{2}\sum_s Z_\theta(s+S,C;R)(-1)^{\mathrm{Arf}[s\cdot\rho]}. \tag{3.50}$$

This partition function is not modular invariant; it depends explicitly on the background spin structure $\rho$. As a sanity check, we set $S = C = 0$ and compute the partition function with spin structure $\rho = AA$. We have

$$
\begin{aligned}
Z_{\text{Dirac}}(0,0;AA;R) \quad &= \frac{1}{2}\sum_s Z_\theta(s,0;R)(-1)^{s_0 s_1} \\
&= \frac{1}{2|\eta|^2}\sum_s\sum_{n\in\mathbf{Z}}\sum_{w\in\mathbf{Z}+\frac{s_1}{2}}(-1)^{(n+s_1)s_0}q^{\frac{1}{2}(n/R+wR/2)^2}\bar{q}^{\frac{1}{2}(n/R-wR/2)^2} \\
&= \frac{1}{|\eta|^2}\sum_{m,\bar{m}\in\mathbf{Z}}q^{\frac{1}{8}(m(2/R+R/2)+\bar{m}(2/R-R/2))^2}\bar{q}^{\frac{1}{8}(\bar{m}(2/R+R/2)+m(2/R-R/2))^2}.
\end{aligned}
$$

Note the explicit symmetry under $R \to 4/R$, corresponding to the map (3.45) between Thirring couplings. At the self-dual point $R = 2$ this reduces to

$$
Z_{\text{Dirac}}(0,0;AA;2) = \left|\frac{1}{\eta}\sum_{m\in\mathbf{Z}}q^{\frac{1}{2}m^2}\right|^2.
$$

This is indeed the partition function of a non-interacting Dirac particle.

### Majorana $\times$ Ising

The Majorana $\times$ Ising theory is constructed by gauging the $\mathbf{Z}_2^C$ charge conjugation symmetry of the Dirac fermion. The action $S_1$ is given in (3.47). By gauging $\mathbf{Z}_2^C$ on both sides of the duality (3.43), we have

$$
S_1(S,C;\rho;R) \quad\longleftrightarrow\quad \int\frac{R^2}{8\pi}(D_{s+S,t}\theta)^2 + i\pi\text{Arf}[s\cdot\rho] + i\pi\int t\cup C.
$$

Thus the partition function can be written as

$$
Z_1(S,C;\rho;R) = \frac{1}{4}\sum_s\sum_t Z_\theta(s+S,t;R)(-1)^{\text{Arf}[s\cdot\rho]+\int t\cup C}. \tag{3.51}
$$

### (Majorana $\times$ Majorana)$/(\mathbf{Z}_2 \times \mathbf{Z}_2)$

The final theory has action $S_2$ given in (3.48). Again, we may follow the gauging in the duality (3.43) to write

$$
S_2(S,C;\rho;R) \quad\longleftrightarrow\quad \int\frac{R^2}{8\pi}(D_{S,t+C}\theta)^2 + i\pi\text{Arf}[t\cdot\rho],
$$

and the partition function is

$$
Z_2(S,C;\rho;R) = \frac{1}{2}\sum_t Z_\theta(S,t+C;R)(-1)^{\text{Arf}[t\cdot\rho]}. \tag{3.52}
$$

Note that this differs from the Dirac partition function (3.50) only by interchanging the two gauge fields coupling to the compact boson.

**Relationships Between Partition Functions**

We can now derive relationships between the theories discussed above. These follow straight-forwardly using the expressions for the partition functions (3.50), (3.51) and (3.52) and our results for the compact boson partition function from Appendix B. The first such identity follows from the $S \leftrightarrow C$ symmetry at $R = \sqrt{2}$, in the form stated in (B.57):

$$
\begin{aligned}
Z_{\text{Dirac}}(S, C; \rho; \sqrt{2}) &= \frac{1}{2} \sum_s Z_\theta(s + S, C; \sqrt{2})(-1)^{\text{Arf}[s \cdot \rho]} \\
&= \frac{1}{2} \sum_s Z_\theta(C, s + S; \sqrt{2})(-1)^{\text{Arf}[s \cdot \rho]} \\
&= Z_2(C, S; \rho; \sqrt{2}).
\end{aligned}
\tag{3.53}
$$

This point is the intersection of the upper horizontal line (the Dirac fermion) and the vertical line $((\text{Majorana} \times \text{Majorana})/(\mathbf{Z}_2 \times \mathbf{Z}_2))$ in Figure 2.

To find the intersection between the lower horizontal line (Majorana × Ising) and the vertical line, we look at

$$
\begin{aligned}
Z_1(S, C; \rho; \sqrt{2}) &= \frac{1}{4} \sum_s \sum_t Z_\theta(s + S, t; \sqrt{2})(-1)^{\text{Arf}[s \cdot \rho] + \int t \cup C} \\
&= \frac{1}{4} \sum_s \sum_t Z_\theta(t, s + S; \sqrt{2})(-1)^{\text{Arf}[s \cdot \rho] + \int t \cup C} \\
&= \frac{1}{2} \sum_s Z_\theta(C, s + S; 2\sqrt{2})(-1)^{\text{Arf}[s \cdot \rho]} \\
&= Z_2(C, S, \rho; 2\sqrt{2}),
\end{aligned}
$$

where the second equality follows from $S \leftrightarrow C$ symmetry (see (B.57)) and the third from the fact that gauging $\mathbf{Z}_2$ without a topological term rescales the radius (see (B.56)). In Figure 2, these intersection points are written in terms of the Thirring couplings using (3.49) rather than the radius.

# Acknowledgements

We're grateful to Matthias Gaberdiel, Shauna Kravec, John McGreevy and Senthil for useful conversations. We especially thank Oscar Randall-Williams for Arf hearted discussions. We are supported by the U.S. Department of Energy under Grant No. DE-SC0011637 and by the STFC consolidated grant ST/P000681/1. DT is a Wolfson Royal Society Research Merit Award holder and is supported by a Simons Investigator Award. CPT is supported by a Junior Research Fellowship at Gonville & Caius College, Cambridge.

# A  Appendix: $\mathbf{Z}_2$ Indices and the Arf Invariant

The mod 2 index of the Dirac operator plays an important role throughout this paper. This index coincides with another object, known as the *Arf invariant*. The purpose of this appendix is to review the connection between these.

A quadratic form on $H_1(X; \mathbf{Z}_2)$ is a function $q : H_1(X, \mathbf{Z}_2) \mapsto \mathbf{Z}_2$ that obeys

$$q(a+b) = q(a) + q(b) + a \cdot b, \tag{A.54}$$

for all $a, b \in H_1(X; \mathbf{Z}_2)$ where this, and all other formulae below, should be understood mod 2. The idea here is that the function $q(a+b)$ has the same algebraic structure as the quadratic function $(a+b)^2$, with the cross-term given by the intersection number of the cycles. Such a function is sometimes said to be a *quadratic refinement* of the intersection number.

Given such a quadratic form, we define the Arf invariant in the following way: pick a symplectic basis $a_i, b_i \in H_1(X; \mathbf{Z}_2)$ with non-trivial intersection numbers $a_i \cdot b_j = \delta_{ij}$, $i, j = 1, \ldots, g$, and write

$$\mathrm{Arf}[q] = \sum_{i=1}^{g} q(a_i) q(b_i).$$

It can be shown the Arf invariant does not depend on the choice of symplectic basis. It provides a classification of the quadratic form up to isomorphism.

It was shown in [57] that there is a 1-1 map between spin structures on $X$ and quadratic forms on $H_1(X; \mathbf{Z}_2)$. The upshot of this argument is as follows: given a spin structure $\rho$, we define the quadratic form $q_\rho(a_i) = 0$ if the spin structure is anti-periodic around $a_i$, and $q_\rho(a_i) = 1$ if the spin structure is periodic around $a_i$, with the same definition for $q_\rho(b_i)$. Furthermore, we set $q_\rho(\circlearrowleft) = 0$ for the homologically trivial curve $\circlearrowleft$, reflecting the fact that a trivial curve is viewed as a $2\pi$ rotation, and so naturally corresponds to anti-periodic boundary conditions. The quadratic formula (A.54) then dictates the extension to other cycles. To see that this gives the expected result for a torus, note that if we go around the cycle $a + b$ then $q_\rho(a+b) = 1$ if and only if $a$ and $b$ are both periodic or both anti-periodic.

To see that life is not as quite as simple as we described above, note that if we take two non-intersecting cycles $a_1$ and $a_2$, each of which are periodic so $q_\rho(a_i) = 1$, then $q_\rho(a_1 + a_2) = 0$ which is anti-periodic. This reflects the fact that spin structures make statements about parallel transport along *framed* curves and such transport around the curve $a_1 + a_2$ involves a $2\pi$ rotation. Further details can be found in [57].

Now we have a correspondence between spin structures $\rho$ and quadratic forms $q_\rho$, it is natural to investigate Arf invariant of $q_\rho$. A classic result of Atiyah [56] shows that the Arf invariant coincides with the mod 2 index of the chiral Dirac operator,

$$\mathrm{Arf}[q_\rho] = \mathcal{I}[\rho]. \tag{A.55}$$

We can easily check this for the torus: in the canonical $a, b$ basis, we see that $\mathrm{Arf}[q_\rho] = q_\rho(a) q_\rho(b) = 1$ if and only if $\rho$ is periodic around both $a, b$. This matches the behaviour of the index, where only the Ramond-Ramond sector contains a zero mode.

One can also construct a Poincaré dual version of the quadratic refinement. Given a spin structure $\rho$, we define the map $Q_\rho : H^1(X; \mathbf{Z}_2) \mapsto \mathbf{Z}_2$ as

$$Q_\rho(s) = \mathcal{I}[s \cdot \rho] + \mathcal{I}[\rho] = \mathrm{Arf}[q_{s \cdot \rho}] + \mathrm{Arf}[q_\rho].$$

Here $q_{s \cdot \rho}$ is a quadratic form on $H_1(X; \mathbf{Z}_2)$ defined by $q_{s \cdot \rho}(a_i) = s(a_i) + q_\rho(a_i)$ (and similarly for cycles $b_i$). The fact this is quadratic follows from the fact that $s \in H^1(X; \mathbf{Z}_2)$ provides a linear map $s : H_1(X; \mathbf{Z}_2) \mapsto \mathbf{Z}_2$.

The function $Q$ is a quadratic function on $s$; this follows from the fact that the Arf invariant is itself quadratic. The addition of the extra term $\mathrm{Arf}[q_\rho]$ ensures that $Q_\rho(0) = 0$. This reflects

the fact that while the space of spin structures is affine, meaning there is no sense in which there is preferred origin, the space of $\mathbf{Z}_2$ gauge fields $s \in H_1(X; \mathbf{Z}_2)$ does have a natural origin. We have chosen to set the root of the quadratic function accordingly.

The quadratic form $Q_\rho$ provides a quadratic refinement of the cup product $s \cup t$. To see this, we use the definition

$$
\begin{aligned}
Q_\rho(s) &= \sum_{i=1}^{g} \big[ s(a_i) + q_\rho(a_i) \big] \big[ s(a_i) + q_\rho(b_i) \big] + q_\rho(a_i) q_\rho(b_i) \\
&= \sum_i s(a_i) s(b_i) + q_\rho(a_i) s(b_i) + s(a_i) q_\rho(b_i).
\end{aligned}
$$

From this, we have

$$
Q_\rho(s+t) - Q_\rho(s) - Q_\rho(t) = \sum_{i=1}^{g} s(a_i) t(b_i) + s(b_i) t(a_i) = \int s \cup t.
$$

Translating back to the mod 2 index using (A.55), we reproduce the identity (2.5),

$$
\mathcal{I}[(s+t) \cdot \rho] = \mathcal{I}[s \cdot \rho] + \mathcal{I}[t \cdot \rho] + \mathcal{I}[\rho] + \int s \cup t.
$$

# B  Appendix: The Compact Boson Partition Function

In this appendix, we review some well known results for a compact boson $\theta$ on a torus. In particular, we wish to keep track of the quantum numbers of states under the discrete symmetries

$$
\mathbf{Z}_2^S : \theta \to \theta + \pi \quad \text{and} \quad \mathbf{Z}_2^C : \theta \to -\theta.
$$

We introduce background $\mathbf{Z}_2$ gauge fields for each of these symmetries. The partition function is then given by

$$
Z_\theta(S, C; R) = \int \mathcal{D}\theta \exp\left[ -\int \frac{R^2}{8\pi} (D_{S,C}\theta)^2 \right].
$$

Note that we have explicitly denoted background fields for both $S$ and $C$ on the compact boson. (In the main text, we mostly suppressed the background field $C$ in equations like (3.33) to avoid clutter.) It is convenient to write $S = (S_0, S_1)$ and $C = (C_0, C_1)$ corresponding to the two cycles on the torus. Each of these components denotes the holonomy around the corresponding cycle, and has value 0 or 1. In the expansion of $Z(S, C)$, this means that $S$-odd operators appear with a $(-1)^{S_0}$ coefficient, and the $S$-twisted sector has $S_1 = 1$.

There is always a part of the spectrum associated with a non-compact boson. These states carry only $C$ charge. We introduce the modular parameter $\tau$ and define $q = \exp(2\pi i \tau)$. This contribution from the non-compact boson is then given by

$$
Z_0(C) = (q\bar{q})^{-1/24} \prod_{r=1}^{\infty} \frac{1}{1 - (-1)^{C_0} q^{r - C_1/2}} \frac{1}{1 - (-1)^{C_0} \bar{q}^{r - C_1/2}}.
$$

Note that each mode is odd under $C$, and carries integer (half-integer) momentum in the $C$-untwisted ($C$-twisted) sector. We have included the appropriate overall factors of $q, \bar{q}$. Writing $C = (C_0, C_1)$, the resulting products can be expressed in terms of $\vartheta_i(\tau)$ and $\eta(\tau)$ functions as

$$
Z_0(0,0) = \frac{1}{|\eta|^2}, \quad Z_0(1,0) = \left| \frac{2\eta}{\vartheta_2} \right|, \quad Z_0(0,1) = (q\bar{q})^{-1/16} \left| \frac{\eta}{\vartheta_4} \right|, \quad Z_0(1,1) = (q\bar{q})^{-1/16} \left| \frac{\eta}{\vartheta_3} \right|.
$$

The global structure is a little more subtle. It will prove useful to consider $C_1 = 0$ and $C_1 = 1$ in turn.

In the $C_1 = 0$ sector, we have the spectrum of a compact boson with momentum $n$ and winding $w$. Note that we have half-integer windings $w \in \mathbf{Z} + \frac{1}{2}$ in the $S_1 = 1$ sector, because we identify $\theta(x+2\pi) \equiv \theta(x)+\pi \pmod{2\pi}$ in this twisted sector. Modes with odd momentum $n$ are $S_0$ odd. Meanwhile, $C$ relates $(n, w) \leftrightarrow (-n, -w)$ and so there is one $C$ even and one $C$ odd linear combination of these objects, except for the $n = w = 0$ state which is $C$ even. The corresponding contribution to the partition function is captured by the function

$$
\begin{aligned}
f(S, C; R) \quad &= \frac{1}{2}\left(1 + (-1)^{C_0}\right) \sum_{n \in \mathbf{Z}} \sum_{w \in \mathbf{Z} + \frac{S_1}{2}} (-1)^{nS_0} q^{\frac{1}{2}(n/R + wR/2)^2} \bar{q}^{\frac{1}{2}(n/R - wR/2)^2} \\
&\quad + \frac{1}{4}\left(1 - (-1)^{C_0}\right)\left(1 + (-1)^{S_1}\right).
\end{aligned}
$$

Meanwhile, in the $C_1 = 1$ sector, we always have two ground states. In the $S_1 = 0$ sector, these are associated to twist operators at $\theta = 0$ and $\theta = \pi$. In the $S_1 = 1$ sector, instead the twist operators sit at $\theta = -\pi/2$ and $\theta = \pi/2$. In all cases, the twist operators have dimensions such that they contribute a factor of $(q\bar{q})^{1/16}$, but in the former case they are both $C$-even whilst in the latter case one combination is $C$-odd.

Putting these together, gives the expression for the partition function

$$
Z_\theta(S, C; R) = Z_0(C)\left[ \frac{1}{2}\left(1 + (-1)^{C_1}\right) f(S, C; R) + \frac{1}{2}\left(1 - (-1)^{C_1}\right)(q\bar{q})^{1/16}\left(1 + (-1)^{S_0 + S_1 C_0}\right) \right].
$$

Armed with this expression, one can compute the partition function of the various other theories obtained by gauging $S$ and $C$. Before doing so, however, it is useful to establish some key properties of the partition function.

First, T-duality. This is the transformation $R \mapsto 2/R$. Importantly, $Z_\theta(S, C; R) \neq Z_\theta(S, C; 2/R)$. This reflects the mixed anomaly discussed in the main text. However, we do have

$$
Z_\theta(0, C; R) = Z_\theta(0, C; 2/R).
$$

There is a second, related identity obeyed by the partition function:

$$
\frac{1}{2} \sum_s (-1)^{\int s \cup S} Z_\theta(s, C; R) = Z_\theta(S, C; 4/R). \tag{B.56}
$$

This corresponds to the fact that if we gauge the $\mathbf{Z}_2^S$ shift symmetry $\theta \to \theta + \pi$, then we obtain a theory of a new scalar at the radius $R/2$, but now $S$ couples to the shift of the dual scalar instead. This is precisely the fact that we used in Section 3.3 to show that the $R = 2$ theory describes Dirac/$\mathbf{Z}_2$. This also confirms the unfamiliar $R \mapsto 4/R$ T-duality map (3.44) for a compact boson coupled to a $\mathbf{Z}_2$ gauge field.

Finally, at the self-dual point $R = \sqrt{2}$, the partition function exhibits knowledge of the enhanced $SU(2)$ symmetry. This is because both $\mathbf{Z}_2^C$ and $\mathbf{Z}_2^S$ correspond to rotations by $\pi$ within different $U(1)$ subgroups of $SU(2)$ [61]. This manifests itself in the values of the partition function $Z_\theta(S, C; \sqrt{2})$ at the self-dual point:

| $\diagdown\ {}^S$ $C$ | $(0,0)$ | $(1,0)$ | $(0,1)$ | $(1,1)$ |
|---|---|---|---|---|
| $(0,0)$ | $\left\|\frac{\vartheta_3(2\tau)}{\eta}\right\|^2 + \left\|\frac{\vartheta_2(2\tau)}{\eta}\right\|^2$ | $\left\|\frac{2\eta}{\vartheta_2(\tau)}\right\|$ | $\left\|\frac{2\eta}{\vartheta_4(\tau)}\right\|$ | $\left\|\frac{2\eta}{\vartheta_3(\tau)}\right\|$ |
| $(1,0)$ | $\left\|\frac{2\eta}{\vartheta_2(\tau)}\right\|$ | $\left\|\frac{2\eta}{\vartheta_2(\tau)}\right\|$ | $0$ | $0$ |
| $(0,1)$ | $\left\|\frac{2\eta}{\vartheta_4(\tau)}\right\|$ | $0$ | $\left\|\frac{2\eta}{\vartheta_4(\tau)}\right\|$ | $0$ |
| $(1,1)$ | $\left\|\frac{2\eta}{\vartheta_3(\tau)}\right\|$ | $0$ | $0$ | $\left\|\frac{2\eta}{\vartheta_3(\tau)}\right\|$ |

We see that one important consequence of the $SU(2)$ symmetry is that, at this point, the two $\mathbf{Z}_2$ symmetries coupled to $S, C$ are equivalent:

$$Z_\theta(S, C; \sqrt{2}) = Z_\theta(C, S; \sqrt{2}). \tag{B.57}$$

# C Appendix: The Dimensional Reduction from 3d to 2d

As we stressed in the introduction, the web of dualities in 2d based on $\mathbf{Z}_2$ gauge fields, is reminiscent of the web of dualities in 3d based on $U(1)$ gauge fields. The Arf invariant in 2d plays a role similar to the Chern-Simons term in 3d. It is natural to ask whether there is a closer link between the two.

This question was partially addressed in [38]. In that paper, we started with the 3d duality

$$\text{free Dirac fermion} \quad \longleftrightarrow \quad U(1)_1 \text{ coupled to XY critical point}$$

and asked what becomes of this duality when compactified down to 2d. Assuming we take periodic boundary conditions around the circle, the free fermion clearly descends to a free Dirac fermion in 2d. In [38], we argued that that the $U(1)_1$ Chern-Simons theory descends to a compact boson at radius $R^2 = 4$.

However, as we discussed in Section 3, this is not quite the right answer: a free Dirac fermion in 2d is dual to a compact boson coupled to a $\mathbf{Z}_2$ gauge field, as in (3.43). In this appendix, we explain how this $\mathbf{Z}_2$ gauge field emerges upon compactification.

The key to understanding this is the dependence of the Chern-Simons term on the background spin structure. We start with the 3d $U(1)_1$ Chern-Simons term

$$S_{CS} = \frac{i}{4\pi} \int_M a \wedge \mathrm{d}a, \tag{C.58}$$

where $M$ is a 3-manifold. As explained in Appendix A of [2], despite appearances this theory depends on the spin structure of $M$. To see this, it is simplest to rewrite the Chern-Simons action as an integral over a 4-manifold $Y$ with boundary $M$,

$$S_{CS} = \frac{i}{4\pi} \int_Y f \wedge f.$$

The spin structure on $M$ should extend to a spin structure on $Y$. This puts restrictions on what 4-manifolds are allowed.

For example, take the 3-manifold to be $M = \mathbf{T}^3 = \mathbf{T}^2 \times \mathbf{S}^1$. If we put anti-periodic boundary conditions on the $\mathbf{S}^1$, then we can construct a 4d manifold by simply filling in $\mathbf{S}^1$ to get a

disc. But with periodic boundary conditions, we have to do something more complicated. The authors of [2] show that if we put a single unit of flux on the $\mathbf{T}^2$, so

$$n_{\text{flux}} = \frac{1}{2\pi} \int_{\mathbf{T}^2} f = 1, \tag{C.59}$$

then the Chern-Simons partition function is $Z_{CS} = 1$ when we have anti-periodic boundary conditions on $\mathbf{S}^1$ and $Z_{CS} = -1$ when we have periodic boundary conditions on $\mathbf{S}^1$.

Now consider the dimensional reduction of the Chern-Simons theory. Again, we take the 3d-manifold to be $M = \mathbf{T}^3 = \mathbf{T}^2 \times \mathbf{S}^1$ with $\mathbf{S}^1$ the Euclidean time direction. We will reduce on a spatial circle in the $\mathbf{T}^2$. The monopole operator in 3d, which we write as $e^{i\theta}$ creates a unit of flux on $\mathbf{T}^2$. However, in a Chern-Simons theory, such monopole operators are not gauge invariant, and must be dressed with a Wilson line. The resulting object is a fermion, which provides another way to understand why the $U(1)_1$ Chern-Simons theory must, ultimately, depend on the spin-structure.

When we reduce on the spatial circle, the low-energy theory must include a term mimicking the above dependence on the spin structure. After compactification, the low-energy degree of freedom is the Wilson line

$$\tilde{\theta} = \frac{1}{2\pi} \int_{\mathbf{S}^1 \subset \mathbf{T}^2} a,$$

where large gauge transformations ensure that $\tilde{\theta} \in [0, 2\pi)$. The presence of flux in the 3d theory is captured by the winding of $\tilde{\theta}$ in the effective 2d theory. Specifically, the flux (C.59) on the original spatial $\mathbf{T}^2$ becomes a winding of $\tilde{\theta}$ around the remaining cycle of the torus:

$$n_{\text{flux}} = \frac{1}{2\pi} \int_{\mathbf{S}^1} d\tilde{\theta}.$$

We need a term in the low-energy effective action which count this winding mod 2 and, if the winding is odd, returns a sign for the partition function $Z = \pm 1$ depending on the spin structure in the temporal direction. Furthermore, the action should be (locally) rotationally invariant, so that the sign of the partition function also depends on a combination of the winding along the temporal $\mathbf{S}^1$ and the spatial spin structure.

It is simple to write down an effective action which achieves this. The low-energy effective action arising from the Chern-Simons term (C.58) is

$$S_{2d} = \frac{i}{2\pi} \int_{\mathbf{T}^2} \tilde{\theta} \, da + i\pi \, \text{Arf}[s \cdot \rho] + \frac{i}{2} \int_{\mathbf{T}^2} s \cup d\tilde{\theta}, \tag{C.60}$$

where the new $\mathbf{Z}_2$ gauge field $s$ is designed to couple the winding mod 2 to the spin structure $\rho$. Including this effect in the dimensional reduction described in [38] results in the correct 2d bosonization duality as described in Section 3.

To see why this is the correct dimensional reduction is slightly more involved. First, we should determine how to compute $S_{CS}$ for a general flux, rather than restricting to (C.59). To do this, first note that integrating $\frac{1}{2\pi} \int da$ around an arbitrary 2-cycle in $M$ gives a quantized winding number, and hence it can be measured mod 2 to give an element of $H^2(M; \mathbf{Z}_2)$. We define the Poincare dual of this to be

$$D\left[\frac{1}{2\pi} \int da \bmod 2\right] \in H_1(M; \mathbf{Z}_2).$$

Then, as we discussed in Appendix A, we can evaluate the spin structure on this element by using the quadratic form $q_\rho : H_1(M; \mathbf{Z}_2) \to \mathbf{Z}_2$. We can write down the topological terms in the action

$$S_{\text{top}} = i\pi q_\rho \left( D \left[ \frac{1}{2\pi} \int \mathrm{d}a \bmod 2 \right] \right). \tag{C.61}$$

As a simple example, if there is flux around $\mathbf{T}^2 \subset \mathbf{T}^3$, then the Poincaré dual gives back the homology element corresponding to the remaining circle. Then we get $Z_{CS} = +1$ or $-1$ for antiperiodic and periodic boundary conditions respectively, as we found above.

Now if we write $M = X \times \mathbf{S}^1$, and impose periodic boundary conditions around the $\mathbf{S}^1$, then the expression (C.61) depends only upon fluxes around cycles involving the $\mathbf{S}^1$ factor. As discussed above, this is simply the winding of the Wilson line around the cycles of $X$, which we write as $t = \frac{1}{2\pi} \int \mathrm{d}\tilde{\theta} \bmod 2 \in H^1(X; \mathbf{Z}_2)$. After a short computation one finds

$$q_\rho(D(t)) = \text{Arf}[t \cdot \rho] + \text{Arf}[\rho] \equiv \text{Arf}[s \cdot \rho] + \int s \cup t, \tag{C.62}$$

and hence (C.60) descends naturally from the 3d spin-TQFT.

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
