# Peer review of "A Web of 2d Dualities: ${\bf Z}_2$ Gauge Fields and Arf Invariants"

_SciPost Physics, doi:SciPost Phys. 7, 007 (2019)_

## Round 2 · Referee Report · Anonymous · 2019-5-9

Strengths

1- A careful and thorough exposition of 2d dualities connecting a family of familiar CFTs
2- Spells out how a seed duality generates the rest through discrete gauging (orbifold)
3- Interesting connections to the 3d duality web
4 -The paper is very well-written and self-contained

Weaknesses

1- The analysis in the paper focused on the simplest (spin)-CFTs in 2d. It would be good to comment on how the analysis generalizes to 1. general interacting 2d (spin)-CFTs, 2. gauging general discrete symmetries, 3. what are potential limitations.

2- As was mentioned in the opening paragraph in the introduction, apart from gauging global symmetries, another way to generate additional dualities from a seed duality is to compactify a dual pair on compact manifolds. I was hoping the authors would expand on this point, i.e. making a connection between the well-known 3d duality web with the 2d web they identified here. Unfortunately the authors didn't seem to come back to this point in the rest of the paper.

Report

The paper revisited familar (spin/fermionic)-CFTs in two spacetime dimensions that describe free bosons and free fermions as well as their Z2 orbifolds. The authors presented a nice story about how a single seed duality: the bosonization duality between the Majorana fermion spin-CFT and a certain Z2 orbifold of the Ising CFT, generates a duality web, via Z2 orbifolding and shifting around topological counter-terms. The authors achieved this by carefully spelling out the statement of the seed bosonization duality: including couplings to background gauge fields, spin structures as well as topological counter-terms such as the discrete cohomology classes and the Arf invariant. The authors explained very explicitly, how all these ingredients are crucial to obtain a matching between the two sides. Interestingly, this allowed the authors to relate the familar Krammers-Wannier duality for Ising model via bosonization, to a Z2 chiral rotation of the Majorana fermion.
The authors went on to consider dualities that involve a Dirac fermion and its Z2 orbifold cousins, and made the connection to T-duality via bosonization. In particular they explained in what sense the Dirac point on the c=1 CFT moduli space is self-dual. Finally the authors presented a nice picture of the c=1 moduli space for fermionic/spin-CFTs which is new and has richer features than the ordinary bosonic c=1 moduli space. They used the dualities they've identified in the paper to predict how different branches of the moduli space intersect.

The paper is very well-written, thorough in analysis, and a pleasure to read. Although many of the ideas were known (scattered in the literature), the authors made an effort to piece them together into a nice self-contained story. Therefore, I recommend this paper for publication after the authors address the requested changes and the minor complaints in the "Weaknesses" section.

Requested changes

1- In equation (2.1), the Majorana conjugate \bar\chi should be defined explicitly to avoid confusions. Especially since the authors switches between Euclidean and Lorenzian signatures in the paper.
2- In the unnumbered equation above (2.20), \sigma^4 should come with + sign
3- pg 14, 2nd paragraph, S_{eff} is i\pi \int t \cup S (not s\cup S)
4- Below (3.15), it was stated that the chiral fermions \psi_{L,R} do not exist in the bosonic theory. To be more precise, they exist as non-local operators, i.e. end points of Z2 symmetry defects.
5- In the unnumbered equation above (3.19), the dynamical Z2 gauge field "t" in the last line should be "v" instead.
6- In equation (3.21), S_{Dirac} should be specified to have C=0 (charge conjugation Z2 gauge field turned off)

  • validity: high
  • significance: high
  • originality: good
  • clarity: top
  • formatting: excellent
  • grammar: perfect

Author:  David Tong  on 2019-05-24  [id 526]

(in reply to Report 1 on 2019-05-09)
Category:
correction

Thank you very much to the referee for such a careful reading of the manuscript.

We will fix all the suggested typos. The question about the relation to 3d dualities is a good one. We actually wrote a paper last year (arXiv:1805.00941) which explains how certain 3d and 2d dualities are related, but we missed the subtleties associated to spin structures that we address in this paper. We will add an extended appendix to the present paper explaining how these subtleties manifest themselves in the reduction from 3d. (The essence is that $U(1)_1$ CS theory depends on the spin structure and this, in turn, results in an Arf invariant term arising after dimensional reduction.)

We will resubmit the manuscript when we hear from the editor.

Thanks again! David

---

## Round 2 · Referee Report · Anonymous · 2019-6-7

Strengths

1. A very clear and pedagogical review of known dualities of 2d CFTs including spin structure dependence.

2. New results explaining how dualities of c=1/2 and c=1 theories fit together. This is probably known to some experts, but as far as I know has not appeared in print.

2. New results concerning the space of c=1 CFTs with spin structure. Again, this might be known to CFT experts, but has not appeared in print.

Report

While neither the approach nor the subject matter are very original, the paper fills a gap in the CFT literature.

---

## Round 3 · Author Response

This resubmission includes all the changes listed in the reply to the first referee.

---

## Round 3 · List of Changes

A number of typos, pointed out by the first referee, have been fixed. We also added a new appendix in response to this referee, explaining how the Arf invariant emerges upon reduction from 3d to 2d.

---

## Editorial Decision

published